# Dimensionality-reduction techniques for complex mass spectrometric datasets: application to laboratory atmospheric organic oxidation experiments

Abigail R. Koss[1*], Manjula R. Canagaratna[2], Alexander Zaytsev[3], Jordan E. Krechmer[2], Martin Breitenlechner[3], Kevin Nihill[1], Christopher Lim[1], James C. Rowe[1], Joseph R. Roscioli[2], Frank N. Keutsch[3], Jesse H. Kroll[1]

[1] Massachusetts Institute of Technology, Department of Civil and Environmental Engineering, Cambridge, MA

[2] Aerodyne Research Incorporated, Billerica, MA

[3] Harvard University, Paulson School of Engineering and Applied Sciences, Cambridge, MA

* Now at Tofwerk USA, Boulder, CO

*Correspondence to*: Abigail Koss (abigail.r.koss@gmail.com)

**Abstract.**

Oxidation of organic compounds in the atmosphere produces an immensely complex mixture of product species, posing a challenge both for their measurement in laboratory studies and their inclusion in air quality and climate models. Mass spectrometry techniques can measure thousands of these species, giving insight into these chemical processes, but the data sets themselves are highly complex. Data reduction techniques that group compounds in a chemically and kinetically meaningful way provide a route to simplify the chemistry of these systems, but have not been systematically investigated. Here we evaluate three approaches to reducing the dimensionality of oxidation systems measured in an environmental chamber: positive matrix factorization (PMF), hierarchical clustering analysis (HCA), and a parameterization to describe kinetics in terms of multigenerational chemistry (gamma kinetics parameterization, GKP). The evaluation is implemented by means of two data sets: synthetic data consisting of a three-generation oxidation system with known rate constants, generation numbers, and chemical pathways; and the measured products of OH-initiated oxidation of a substituted aromatic compound in a chamber experiment. We find that PMF accounts for changes in the average composition of all products during specific periods of time, but does not sort compounds into generations or by another reproducible chemical

process. HCA, on the other hand, can identify major groups of ions and patterns of behavior, and maintains bulk chemical properties like carbon oxidation state that can be useful for modeling. The continuum of kinetic behavior observed in a typical chamber experiment can be parameterized by fitting species' time traces to the GKP, which approximates the chemistry as a linear, first-order kinetic system. Fitted parameters for each species are the

number of reaction steps with OH needed to produce the species (the generation) and an effective kinetic rate constant that describes the formation and loss rates of the species. The thousands of species detected in a typical laboratory chamber experiment can be organized into a much smaller number (10-30) of groups, each of which has characteristic chemical composition and kinetic behavior. This quantitative relationship between chemical and kinetic characteristics, and the significant reduction in the complexity of the system, provide an approach to

understanding broad patterns of behavior in oxidation systems and could be exploited for mechanism development and atmospheric chemistry modeling.

## Introduction

Air quality and climate change are major threats to the quality of millions of human lives across the globe (IPCC, 2014; Landrigan et al., 2018). An important scientific component of both topics is the photooxidation

chemistry of organic compounds in the atmosphere, which can lead to the formation of ozone and fine particulate matter, both of which can affect the radiative budget of the atmosphere and can harm human health. A detailed understanding of this chemistry is necessary to predict and mitigate these effects. However, this is challenging because of the diversity and number of species involved. Gas-phase organic compounds emitted directly into the atmosphere have a wide range of functionality and reactivity, and oxidation of these precursors by $O_3$, OH, or

$NO_3$ can further functionalize or fragment the molecules. The number and diversity of the product species increases with the number of generations of reaction, and key properties of these product species, such as volatility, reactivity, and concentration, can vary over orders of magnitude (Glasius and Goldstein, 2016; Goldstein and Galbally, 2007).

This complexity presents several challenges. In order to fully characterize oxidation of organic compounds,

analytical techniques must be able to detect hundreds to thousands of individual species and accommodate the diversity of functionality and concentration. Advances in instrumentation, especially high-resolution time-of-flight chemical ionization mass spectrometry (CIMS), have enabled detection of a large number of oxidation products in chamber and field experiments. CIMS involves the introduction of a reagent ion, which then reacts with the analyte, forming product ions that are detected with mass spectrometry. Chemical selectivity can be

achieved through choice of the reagent ion, and fast, online measurement of air samples is possible. CIMS instruments with high mass resolution (maximum FWHM m/$\Delta$m >3000) can unambiguously determine the elemental composition of most detected ions with *m/z* less than 200, and the elemental composition of ions with *m/z* >200 can usually be determined with some certainty (Junninen et al., 2010). The analytical capability of atmospheric CIMS instrumentation is rapidly improving, and modern instruments can have sensitivities on the

order of 10000 cps ppbv$^{-1}$ and resolution greater than 10000 m/$\Delta$m (Breitenlechner et al., 2017; Krechmer et al., 2018), allowing the measurement of hundreds to thousands of species on a rapid time base (Isaacman-VanWertz et al., 2017; Müller et al., 2012).

    While this represents a major advance in our ability to detect and characterize trace atmospheric chemical components, these large data sets can be difficult and time-consuming to interpret, and it is not clear how the full

information content from thousands of ions can be best used. Further, secondary ion processes, such as cluster formation or ion fragmentation, can occur within the mass spectrometer, complicating the mass spectra, and different CIMS techniques have differing chemical specificities that can be hard to predict. Data analysis techniques are therefore needed to efficiently reduce the amount of data to more manageable and interpretable sizes. Further, the interpretation of these measurements in terms of chemical mechanisms is often not

straightforward. Most laboratory studies use CIMS measurements to support, refute, or suggest new chemical mechanisms; this is typically done by hand, focusing on several key species of interest. Data analysis techniques that allow for the extraction of useful chemical and mechanistic information from entire mass spectra are valuable and necessary, but have not been systematically explored.

    Simplification is also needed to incorporate oxidation chemistry into climate and air quality models. Large-

scale regional and global models (e.g., chemical transport models, earth system models) cannot currently incorporate a high level of chemical detail. Photochemical mechanisms commonly used to incorporate chemistry into regional and global models typically include 30-200 species and 100-400 reactions (Brown-Steiner et al., 2018; Jimenez et al., 2003), which is much lower than the number of product species from individual precursors included in explicit chemistry mechanisms such as the Master Chemical Mechanism (300-1000+ product species,

e.g. Bloss et al., 2005; Jenkin et al., 2003; Saunders et al., 2003) or GECKO-A (~$10^5$ species, Aumont et al., 2005). In order to reduce the number of species in models, VOCs are represented by groups, or are "lumped," and the choice of lumping criterion can affect the derived ozone, aerosol, and product VOC formation values (Jimenez et al., 2003; Zhang et al., 2012). In gas-phase mechanisms, compounds have been lumped by degree of unsaturation, emission rates, functional groups, or reactivity towards OH (Brown-Steiner et al., 2018; Crassier et

al., 2000; Houweling et al., 1998; Jimenez et al., 2003; Gery et al., 1989; Carter, 1990; Stockwell et al., 1997).

Similarly, secondary organic aerosol formation has been parameterized by lumping organic species by volatility, O:C ratio, number of carbon and oxygen atoms, or polarity, and assigning kinetic properties to each group (Cappa and Wilson, 2012; Donahue et al., 2012; Lane et al., 2008; Pankow and Barsanti, 2009). Lumping schemes could be improved by using laboratory data to define important groups of compounds, and assign experimentally-derived chemical and kinetic properties to each group to act as a surrogate species.


Several methods have been used to categorize mass spectra and to group compounds. We consider two methods previously used to reduce the dimensionality of complex atmospheric chemistry measurements, positive matrix factorization (PMF) and hierarchical clustering analysis (HCA). Both methods have seen substantial use in the simplification and interpretation of field measurements, but have seen far less use in the laboratory, and there has been little exploration of how they can be used to gain useful chemical or mechanistic information from laboratory mass spectrometric datasets. We additionally address a fundamental, underexplored problem in laboratory chamber studies: how to systematically characterize the kinetics of an oxidation system. The systematic characterization is achieved through the gamma kinetics parameterization (GKP) and can be used to group compounds based on similar kinetic properties. The three methods (PMF, HCA, and GKP) have different mathematics but the same goals: to identify groups of compounds, and replace each group with a chemically meaningful surrogate. The three methods are evaluated in terms of the following criteria: whether the resulting surrogates have chemically realistic behavior; whether the surrogates have the same range of chemical properties as the original data set; which subjective choices the researcher needs to make when implementing the method; and what other new information about the system can be learned. We additionally discuss the extent to which different methods agree in their identification of major groups of compounds. The output of these dimensionality-reduction techniques can be used to quickly analyze and interpret chamber experiments, and could be used to reduce the complexity of chemical mechanisms included in models.




## 2 Methods

### 2.1 Data collection


We use two data sets: a synthetic data set describing a simple multigenerational kinetic system, and measurements of the OH-initiated oxidation of 1,2,4-trimethylbenzene in an environmental chamber. The synthetic dataset is useful for evaluating the various dimensionality-reduction schemes used here, because the

reaction rate constants and generation of each species are known exactly. The chamber data demonstrates the application of the data reduction techniques to a real-world system measured with online mass spectrometry.

**2.1.1 Synthetic data set**

A schematic of the simple synthetic kinetic system is shown in Figure 1. The precursor molecule *A* reacts with OH to produce first-generation species (*B*), which in turn reacts with OH to produce second generation (*C*) and further to third-generation species (*D*). Only reactions with OH are considered. The system includes three pathways with differing yields, and each pathway includes a product with a fast, a slow, and an intermediate OH

rate constant. The different rate constants (randomly generated) and yields simulate a range of product behavior. To enable PMF measurements, artificial noise was added to the synthetic data. The noise is normally distributed with a standard deviation proportional to the square root of the signal. The proportionality constant, based on a typical PTR-MS sensitivity of 10,000 counts $ppb^{-1}$ $s^{-1}$, was chosen to generate signal-to-noise ratios between 10 and 100, a reasonable range for chamber experiments.

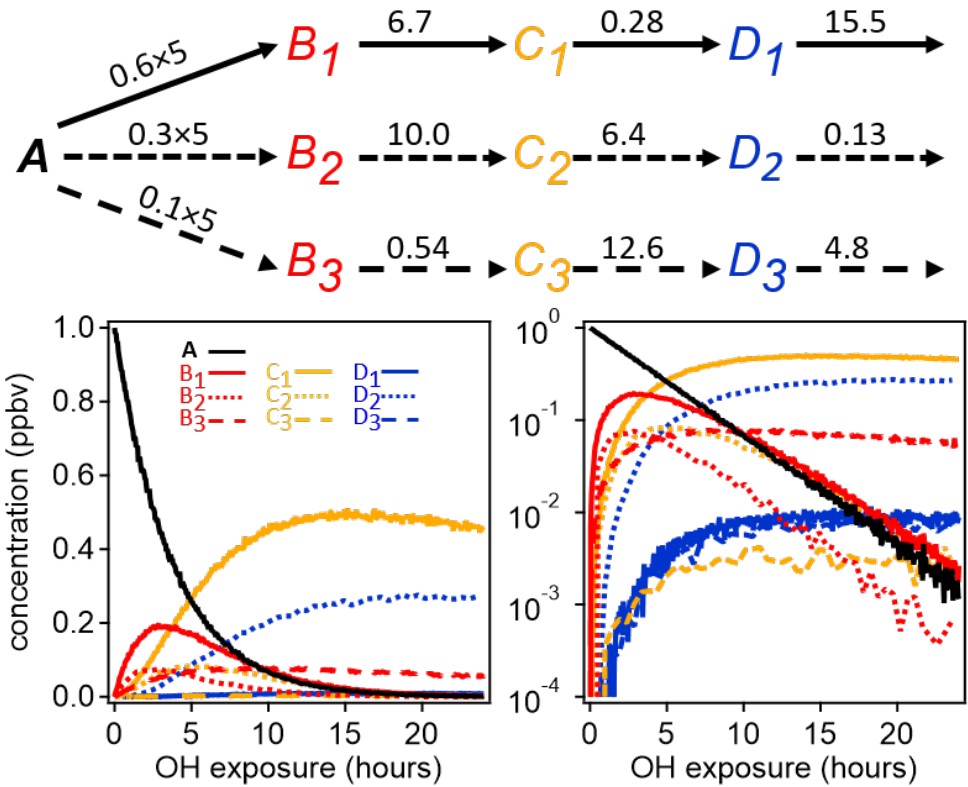

**Figure 1** Schematic of reaction pathways with OH (top) of synthetic data and time series shown with linear and log concentration (bottom: left and right, respectively). Arrows represent a reaction with OH. Reaction rate constants with OH are written above the arrows (units are in $10^{-11}$ cm$^3$ molecule$^{-3}$ s$^{-1}$). Precursor species A reacts at a rate of 5 $\times10^{-11}$ cm$^3$ molecule$^{-3}$ s$^{-1}$ with yields of 0.6, 0.3, and 0.1 for the three pathways, respectively. Products of pathways 1, 2, and 3 are drawn with solid, short-dash, and long-dashed lines, respectively, and the first-, second-, and third-generation products are drawn in red, yellow, and blue. The total OH exposure is equal to 24 hours at an average OH concentration of $1.5\times10^6$ molecule cm$^{-3}$.

### 2.1.2 Chamber oxidation of 1,2,4-trimethylbenzene

An oxidation experiment was conducted in the MIT environmental chamber, which consists of a 7.5m$^3$ PFA enclosure. The chamber conditions were controlled at 20 °C and 2% relative humidity. The chamber is illuminated by forty-eight 40 W blacklights with a 300-400 nm spectrum peaking at 350 nm. During experiments the chamber maintains a constant volume, and clean air is continuously added at a rate equal to the instrument sample flow (15 lpm). Additional details of chamber operation have been previously reported (Hunter et al., 2014).

Dry ammonium sulfate seed (which provide surface area onto which low-volatility vapors can condense) was first added to the chamber to reach a number concentration of $5.7\times10^4$ cm$^{-3}$ (19.7 µg m$^{-3}$). Nitrous acid (HONO, the OH precursor) was added by bubbling clean air through a dropwise addition of $H_2SO_4$ to $NaNO_2$ to reach a concentration of 45 ppbv in the chamber. Several ppbv of an unreactive tracer, hexaflurorobenzene, were added to provide a measure of chamber dilution. Three microliters of neat 1,2,4-trimethylbenzene (SigmaAldrich) were added by injection into a 70°C heated inlet with a flow rate of 15 lpm, resulting in an initial concentration of 69 ppbv in the chamber. The reagents were allowed to mix for 15 minutes, then the experiment was initiated by turning on lights to photolyze nitrous acid and generate OH. Measurements were conducted for seven hours. During this time three additional aliquots of nitrous acid (27 ppbv, 10 ppbv, and 18 ppbv) were added at regularly-spaced intervals to roughly maintain the OH concentration. The OH concentration was determined by fitting a double-exponential function to the measured decrease of 1,2,4-trimethylbenzene, including a known dilution term (determined from hexafluorobenzene dilution) and an OH reaction term. A total atmospheric-equivalent exposure of 16.5 hours (assuming an average atmospheric OH concentration of $1.5\times10^6$ molecule cm$^{-3}$) was achieved.

CO and formaldehyde were measured by tunable infrared laser differential absorption spectroscopy (TILDAS, Aerodyne Research Inc.) Other gas-phase organic species were measured by chemical ionization, followed by analysis with high-resolution time-of-flight (HR-ToF) mass spectrometry. Three chemical ionization mass spectrometry (CIMS) techniques were used: I$^-$ reagent ion, $H_3O^+$ reagent ion, and $NH_4^+$ reagent ion. The I$^-$ CIMS instrument is from Aerodyne Research Inc. and is described by Lee et al. (2014). $H_3O^+$ and $NH_4^+$ CIMS involved proton-transfer-reaction mass-spectrometers with switchable reagent ion chemistry (PTR3-$H_3O^+$ and PTR3-$NH_4^+$, Ionicon Analytik). The PTR3 $H_3O^+$ CIMS and $NH_4^+$ CIMS techniques are described by Breitenlechner et al., 2017 and Zaytsev et al., 2019, respectively. $H_3O^+$ CIMS was also carried out using a second proton-transfer-reaction mass spectrometer (Vocus-2R-PTR, TOFWERK, A.G.), which is described by Krechmer et al., 2018. Total organic aerosol mass was measured using a high-resolution time-of-flight aerosol mass spectrometer (AMS) from Aerodyne Research Inc. (DeCarlo et al., 2006), calibrated with ammonium nitrate and assuming a collection efficiency of 1. Organic aerosol accounted for approximately 2% of the secondary carbon, and individual ion measurements from the AMS are not considered separately. The TILDAS was calibrated directly for CO and formaldehyde. The Vocus-2R-PTR was calibrated directly for 1,2,4-trimethylbenzene and acetone. The PTR3 $H_3O^+$ CIMS was calibrated directly for 15 individual species and an average calibration factor was applied to other species. The PTR3-$NH_4^+$ and I$^-$CIMS were calibrated using a combination of direct calibration and collision-induced-dissociation (Lopez-Hilfiker et al., 2016; Zaytsev et al., 2019). We note however that the calibration of each instrument does not affect any results presented in this work, since the analysis

techniques used examine the time-dependent behavior, and not the absolute concentrations, of the measured species.

Sampling from the chamber to CIMS instruments was designed to reduce inlet losses of compounds as much as possible, within the physical constraints of the chamber. Each instrument used a 3/16" ID PFA Teflon line of 1m or less in length, with a flow of 2 LPM. Inlets extended 10cm into the chamber and no metal fittings were used. The PTR instruments additionally have instrument inlets and ion-molecule-reaction chambers that minimize gas contact with walls (Breitenlechner et al., 2017; Krechmer et al., 2018). In this study, CIMS inlet (including chamber and instrument inlet) loss timescales were 15 seconds or less for test compounds with saturation concentrations between $10^2$ and $10^7$ µg m$^3$ and therefore wall interactions for these species are unlikely to affect the observed kinetics, which occur over tens of minutes (Krechmer et al., 2016).

Chamber background for each measurement was determined from measurements taken prior to precursor injection, which are subtracted from each chamber measurement reported. All measurements were also corrected for dilution by normalizing to the hexafluorobenzene tracer (for gas-phase data) or to measured $(NH_4)_2SO_4$ aerosol seed (for particle-phase data, which also corrects for wall loss and AMS collection efficiency).

Between 1000 and 3000 peaks with variability above background were observed in the mass spectra from each CIMS instrument; these include chemistry-relevant ions related to oxidation products, as well as other ion signals from sources such as instrument ion sources, the hexafluorobenzene dilution tracer, tubing and inlets, and interferences from large neighboring peaks in the mass spectrum (Cubison and Jimenez, 2015). Two data-processing steps were used to identify the chemically relevant ions.

First, the elemental formulas of all ions were determined. With the resolution of the instruments used here (~maximum 10000 m/Δm for Vocus-2R-PTR and PTR3; ~3000 for I$^-$CIMS), elemental composition can become ambiguous at high $m/z$ values. We first assigned all unambiguous peaks, where only one reasonable formula within 10ppm of the peak was possible, beginning with the largest peaks in order to identify and exclude isotopes. Then, we used trends observed in Kendrick mass defect plots to suggest formulas for species expected at higher masses. Remaining peaks (<1% of instrument signal) were assigned the formula with the nearest mass that included C, H, N, and O, had nine or fewer carbon atoms, and had positive, integer double-bond-equivalency (again, beginning with the largest peaks and excluding isotopes). A mass defect plot showing unambiguous ions, and the complete set of ions, is shown in Figure S1.

Second, chemically relevant ions were separated from all other ion signals using hierarchical clustering. Chemically relevant ions are those which result from oxidation products. They are enhanced above background during the oxidation experiment and do not have sudden, stepwise changes, which would indicate an instrument

interference. A difference mass spectrum, which compares the average signal of each ion before chemistry is initiated to the average signal during oxidation, is a simple way to identify relevant ions, but can be misleading for ions with low signal-to-noise ratios or variability unrelated to oxidation chemistry. Hierarchical clustering provides an alternative method, involving the systematic examination of the time-dependent behavior all measured species. Chemically relevant ions exhibit a time dependence that is consistent with chemical kinetics (formation

of the product, often followed by reactive loss) that is different from that of ions not resulting from oxidation. These two classes are clustered separately from each other, enabling the straightforward selection of only chemically relevant ions. The hierarchical clustering algorithm is described in section 2.2.2. An example for the PTR3 $H_3O^+$ mode instrument is shown in Figure S2. This approach was used to identify chemically relevant ions and to exclude all background ions from each CIMS instrument.

Compounds that were measured by more than one instrument, identified as having the same elemental composition (after correction for any reagent ion chemistry) and similar time-series behavior (Pearson's R >0.9), were included only once in the data set with all product species. When selecting compounds measured by more than one instrument, data from PTR-MS instruments, which have the smallest calibration uncertainties, were used first, followed by $I^-$ CIMS and $NH_4^+$ CIMS. In the final, combined data set, approximately half the carbon in

oxidation products was measured by PTR-MS, with about 15% measured each by $I^-$ CIMS, $NH_4^+$ CIMS, and TILDAS, and an additional 2% by AMS. We recognize that there is a great deal of uncertainty associated with calibrating CIMS instrumentation and identifying detected ions. This is an active area of research that we do not attempt to address fully here. Calibration and identification of species measured by more than one instrument do not affect the major conclusions of this paper.

**2.2 Implementation of data simplification tools**

**2.2.1 Positive Matrix Factorization (PMF)**

In atmospheric chemistry, PMF analysis typically involves representing a time series of mass spectra (or other chemical measurements), recorded as a matrix of *m* measurements by *t* time points, as a linear sum of "factors" (Paatero, 1997; Ulbrich et al., 2009; Zhang et al., 2011). Each factor is fixed in chemical composition, but varies

in intensity over time.

PMF analysis of ambient air measurements has in many situations been shown to be robust and meaningful, and has contributed greatly to our understanding of atmospheric and aerosol chemistry. PMF is frequently used for source apportionment and characterization of organic aerosol in field studies, for example, to sort aerosol as

more- or less- oxidized, or from a specific source such as biomass burning (Zhang et al., 2011). PMF is also
frequently applied to VOC measurements in field studies. In this application, each factor indicates a particular
VOC class (which can be associated with a specific source) and its magnitude, which is a powerful tool to support
regulation.

Some aspects of atmospheric chemistry can complicate PMF analysis. Oxidation chemistry during transport
from the source to the measurement location can change the chemical composition, causing a single source to
appear as several factors, or causing oxidized species from several sources to be grouped together, and adding
substantial uncertainty to the derived source profiles (Sauvage et al., 2009; Wang et al., 2013; Yuan et al., 2012).
Factors including oxidation products, described as "secondary" or "long-lived species," or that require correction
for photochemistry have been reported in a number of studies from diverse locations (e.g. Abeleira et al., 2017;
Sarkar et al., 2017; Shao et al., 2016; Stojić et al., 2015), but the interpretation of such factors within the context
of a continually-evolving system is unclear.

Finally, PMF has been applied to measurements of oxidizing chemical systems to greatly reduce the
complexity of the dataset and identify key shifts in chemistry, including aerosol in laboratory experiments (e.g.
Craven et al., 2012; Fortenberry et al., 2018), VOCs in chamber experiments (Rosati et al., 2019), and gas-phase
highly-oxidized molecules in field studies (Massoli et al., 2018; Yan et al., 2016). Therefore, it is important to
understand whether PMF analysis of an oxidizing system returns chemically distinct, reproducible factors that
correspond to a physical or chemical aspect of the system.

The algorithm was implemented using the PMF Evaluation Tool v2.08 (Ulbrich et al., 2009) using the
PMF2 algorithm (Paatero, 2007). We chose this implementation because it is widely used in atmospheric science
and has been optimized for atmospheric chemistry data. Briefly, the algorithm takes as input an m×n matrix of
measured data M, containing n measured compounds at m time points, and a matrix of estimated error (one
standard deviation, σ) for each point in the measured data matrix. The solution for a given number of factors p is
given as an m×p matrix G of factor time series, a p×n matrix F of factor profiles, and a matrix E that contains the
residual (M-GF). F and G are iteratively adjusted to minimize the quality-of-fit parameter Q: :

$$Q = \sum_{i=1}^{m} \sum_{j=1}^{n} \left( e_{ij} / \sigma_{ij} \right)^2$$

where $e_{ij}$ is the residual between the measurement and the PMF reconstruction of compound j at time point i, and
$\sigma_{ij}$ is the estimated error of that measurement.

The factors and their profiles are constrained to be non-negative. The measured data matrix **M** for the
synthetic dataset was constructed using all ten species (precursor plus 9 products) with artificial noise. The

measured data matrix **M** for the chamber dataset was constructed using all measured product species (defined as all chemically-relevant ions from CIMS instruments, plus total organic aerosol, CO, and formaldehyde), after background subtraction, dilution correction, and calibration in units of parts-per-billion carbon (ppbC). Duplicate measurements of individual species from multiple instruments were excluded. Although calibrated data are used here, because PMF operates on the unitless quality-of-fit parameter $Q$, the results are not sensitive to calibration, only to the signal-to-noise ratio of the individual measurements.

Because the precursor compound (1,2,4-trimethylbenzene) has an average intensity an order of magnitude larger than any other species, and therefore a very high signal-to-noise ratio, if it is included in **M** the quality-of-fit parameter $Q$ and the resulting solution are dominated by the precursor. As this is not of interest, the precursor was also excluded in PMF analysis. Data were interpolated to 500 points evenly-spaced with respect to OH exposure (0-16.5 atmospheric-equivalent hours).

The matrix of estimated errors for the synthetic dataset was taken as the standard deviation used to generate the artificial noise. The matrix of estimated errors for the chamber dataset was generated by smoothing the data using a running 20-minute linear best-fit, and subtracting this smoothed data from the original measurement. The standard deviation of the residual within a 20 minute window was determined for each time point. Signal-to-noise ratios for both synthetic and chamber data are shown in Figure S3. The overall relationship between the standard deviation determined for chamber data and the measured concentration is reasonable (Figure S4).

Rotational forcing, which examines linear combinations of possible solutions using the parameter fPeak, was explored through fPeak values between -1 and 1. The selected fPeak was chosen to avoid factor time-series with multiple maxima, which are not physically realistic in the chamber system. Solutions were also explored using different random initialization values, or seeds. No significant differences were found between solutions with random seed values 1-10.

When PMF is used to reduce the complexity of a dataset, the number of factors must be chosen by the researcher, a choice that is inherently subjective. Solutions were explored with one to ten factors for the synthetic dataset and the chamber data.

**2.2.2 Hierarchical Clustering Analysis (HCA)**

A second technique is to group or cluster individual measurements based on the similarity of their behavior over time. While a measurement of a single chemical species can contribute to more than one PMF factor, it can belong to only one cluster. Several approaches to clustering exist. The approach we consider here is agglomerative hierarchical clustering, which describes the degree of similarity between any two measurements

and can be used to sort species into categories of behavior (Bar-Joseph et al., 2001; Müllner, 2011). Hierarchical clustering analysis (HCA) has been used to group aerosol particles based on the similarity between individual mass spectra determined by aerosol mass spectrometers (Marcolli et al., 2006; Murphy et al., 2003; Rebotier and Prather, 2007), describe time-series of thermally-desorbed organics measured by CIMS (Sánchez-López et al., 2014; Sánchez-López et al., 2016), and recently to determine the appropriate number of PMF factors used to analyze PTR-MS data from chamber studies (Rosati et al., 2019). To our knowledge it has not yet been used to group compounds with similar time-varying behaviors to understand chemical transformation in an oxidation system. In this work we show how this technique can be implemented, and assess its ability to reduce the complexity of a dataset while maintaining chemical information.

Agglomerative hierarchical clustering sorts measurements by similar time-series behavior, and displays the relative similarity between measurements. First, all measurements were normalized, so that the time-series behavior could be directly compared despite differences in absolute concentrations or detection efficiencies. Data are noisy, and noise can contribute to the absolute highest point in a time series. To account for this, we normalized data to the average of the 10 points surrounding the highest point in each time series. Then, the distance between each pair of measurements $A$ and $B$ was determined. The distance describes the dissimilarity between any two time series measurements: two identical time series have a distance of zero, and measurements with different time-series behavior have larger distance values. Distance was calculated by summing the differences between the normalized measurement intensities $A$ and $B$ over all time points $t$:

$$d_{AB} = \sum_t abs(A_t - B_t).$$

Other distance metrics are possible, including using a correlation coefficient or the sum of squared residuals. This particular approach was chosen because it resulted in the grouping that was most reproducible and understandable, and least sensitive to outlier points in the time series.

The algorithm begins with the distances between all original measurements. The pair of measurements $s$ and $t$ with the lowest distance value is found, and these two measurements are assigned to a new cluster $u$. The two original measurements $s$ and $t$ are removed from the set, and the new cluster $u$ is added. Then, the distances between the new cluster $u$ and all the remaining measurements are determined. The algorithm then iteratively searches for the next smallest distance value and combines the pair into a new cluster. As the algorithm iterates, new clusters can be formed from two original measurements, from an original measurement and a cluster, or from two clusters. The distance between the new cluster $u$ and any other measurement or cluster in the set $v$, is calculated as the average of the distances between each of the "$n$" individual members of $u$ and "$m$" individual members of $v$, over all points $i$ in cluster $u$ and points $j$ in cluster $v$:

$$d_{uv} = \sum_{i,j} \frac{d(u_i, v_j)}{m \times n}$$

The algorithm continues until only one cluster remains. Clustering was implemented using the open-source scipy.cluster.hierarchy.linkage package (SciPy.org, 2018). The relationships between each of the different measurements and clusters are visualized using a dendrogram.

Compounds must be grouped into a specific number of clusters in order to use HCA to define surrogate

species. The average chemical and kinetic properties of each cluster can be used to define a surrogate species. As with the number of factors from PMF, the number of clusters is subjectively chosen by the researcher. The clusters could be selected by hand, or by choosing a threshold for distance $d_{AB}$ to automatically define clusters. We chose to use a threshold to define the number of clusters, and considered several different values of thresholds that result in different numbers of clusters. The effect of threshold value on the interpretation of the data is discussed in

Section 3.2.

### 2.2.3 Gamma Kinetics Parameterization (GKP)

To date, bulk characterization of oxidation products in photochemical chamber experiments has largely focused on their chemical composition, and not their reactivity or mechanistic relationship. A few studies have derived kinetic information from time-series data (Smith et al., 2009; Wilson et al., 2012), but this has been limited

to aerosol-aging experiments and not to atmospheric oxidation generally. A chamber oxidation experiment with speciated mass spectrometric measurements also contains a great deal of kinetic information, because the rates of formation and decay of each species are measured. In this work we show how the kinetic behavior of any particular measurement can be parameterized using a simple function, the gamma kinetics parameterization (GKP), which describes a system of first-order linear multi-step reactions. The function returns parameters that describe

generation number (how many OH addition steps are needed on average to create the molecule), and reactivity (the relative rates of formation and decay), which are shown to correlate with key chemical characteristics. Grouping by similar kinetic parameters suggests a new, experimentally-derived approach to lumping mechanisms.

A multigeneration reaction system can be described as a linear system of first-order reactions:

$$X_0 \xrightarrow{k_0} X_1 \xrightarrow{k_1} X_2 \xrightarrow{k_2} ... X_m \xrightarrow{k_m} X_{m+1} \xrightarrow{k_{m+1}} ... \quad \text{(Eq. 1)}$$

where $k_i$ is the rate constant and $m$ is the number of reactions needed to produce species $X_m$ (i.e., the generation number). When all $k_i$'s are equal, the series of differential equations that describe the kinetics of Eq. 1 can be solved analytically, with the time dependence of any compound $X_m$ described by:

$$[X_m](t) = a(kt)^m e^{-kt} \text{ (Eq. 2)}$$

where $a$ is a scaling factor that depends on both instrument sensitivity and stoichiometric yield (Smith et al., 2009; Wilson et al., 2012; Zhou and Zhuang, 2007). This function is related to the probability density function of the gamma distribution, a continuous probability distribution that has been previously used in chemistry to characterize protein kinetics (Pogliani et al., 1996; Zhou and Zhuang, 2007).

Oxidation reactions in a chamber experiment can be parameterized as a linear system of reactions, but the reactions between organic compounds and OH are bimolecular. This can be adjusted to a pseudo-first-order system by considering the integrated OH exposure $[OH]\Delta t = \int_0^t [OH]dt$ instead of reaction time $t$. In this case, the observed behavior of an organic compound X that reacts with OH in the chamber can be parameterized by:

$$[X_m](t) = a(k[OH]\Delta t)^m e^{-k[OH]\Delta t} \text{ (Eq. 3)}$$

where $k$ is the second-order rate constant (units of cm$^3$ molecule$^{-1}$ s$^{-1}$), $m$ is the number of reactions with OH needed to produce the compound (generation number), and $[OH]\Delta t$ is the integrated OH exposure (units of molecule s cm$^{-3}$). This parameterization is exact in the situation where all rate constants $k$ in the system are equal, and is an approximation otherwise, in which $k$ is an effective rate constant representing the overall rate of reactions in the pathway.

Figure 2 illustrates how the parameters $a$, $k$, and $m$ relate to the shape of the function described in Equation 3. The parameter $m$ (Figure 2a) returns the generation number and is determined by the curvature of [X] as $[OH]\Delta t \rightarrow 0$ (Zhou and Zhuang, 2007).

Eq. 3 can be fit to time-dependent concentration (or ion intensity) data to return $a$, $k$, and $m$. The fitted value of $m$ can be affected by noise or by fitting to a too-long timestep (Zhou and Zhuang, 2007). The optimum timestep depends on the signal-to-noise ratio of the data and the compound's reaction rate, but can be determined empirically. The fit can also be improved by integrating the data with respect to OH exposure over the experimental time period, and fitting the integrated form of Eq. 3, which reduces random Gaussian noise (Section S1). When all rate constants within a reaction sequence are not identical (which is typically the case), there is no direct analytical relationship between the effective rate constant $k$ (Figure 2b) and the individual rate constants in the pathway. However, the effective rate constant $k$ provides a rough measure of the reactivity of the compound and its precursors. A higher effective $k$ indicates higher formation and/or reaction rates, and is affected by rate-limiting steps. The scaling constant $a$ (Figure 2c) ensures that the returned values of $k$ and $m$ are insensitive to instrument calibration and stoichiometric yields.

Compounds can be grouped by similar $k$ and $m$ in order to reduce the complexity of the dataset. The $k$, $m$, and average chemical properties of the group can be used to define a surrogate species. The choice of the

number of groups and the method of grouping are subjective. GKP could be used alone, by binning compounds
by similar $k$ and $m$, or it could be used in combination with another analysis technique, such as HCA. Several
approaches to using GKP to define surrogate species are discussed in section 3.3.2.

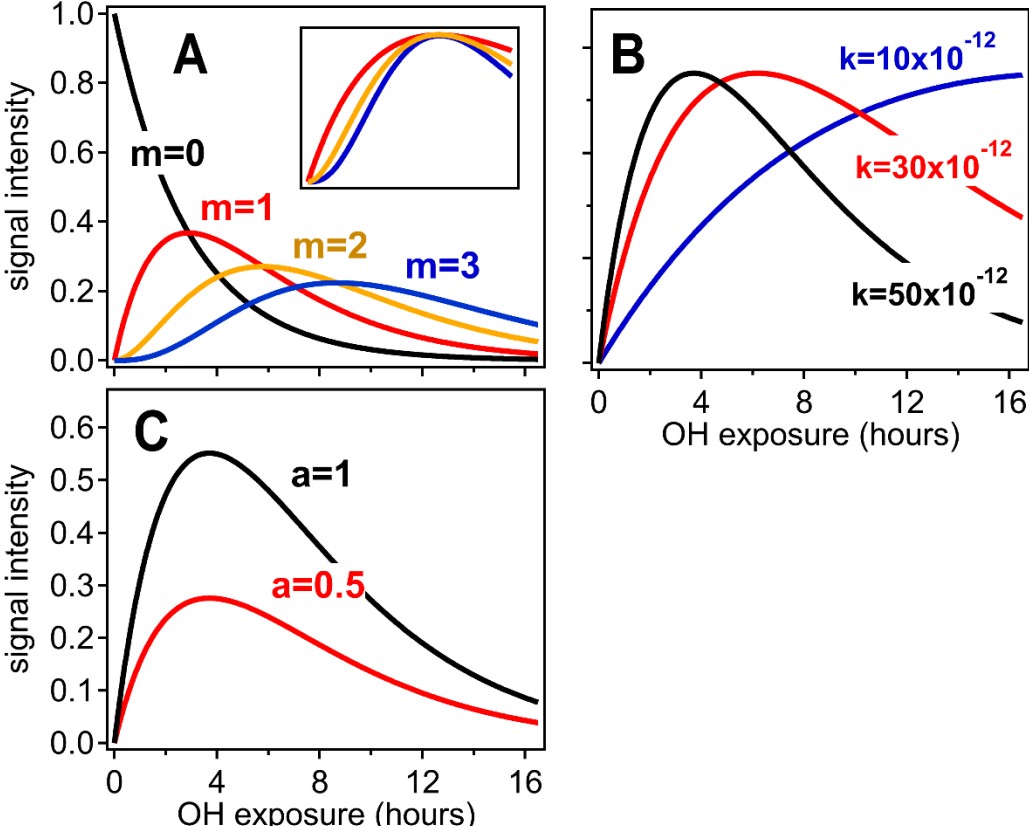

**Figure 2** Illustration of the relationships between the different GKP parameters ($m$, $k$, and $a$) and the time dependence of a
given species, using synthetic data. A. Parameterizations with different generation $m$. In the subpanel, the traces with $m=2$ and
385 $m=3$ have been scaled to allow comparison of the curvature, which differs with generation. B. Parameterizations with different
rate constant $k$. Increasing $k$ does not change the shape of the curve, but causes the maximum to occur at lower OH exposures.
C. Parameterizations with different scaling constant $a$, which changes neither curvature nor location of the maximum, but only
the height of the curve.

**3 Results and discussion**

**3.1 PMF**

**3.1.1 PMF of synthetic data**

A set of PMF solutions for the synthetic data, including 2-10 factors, is shown in the Supplement (Figure S5). The quality of the PMF reconstruction can be evaluated in two ways: the residual between the PMF reconstruction and the original data (lower residual indicates better agreement), and the normalized mutual information (NMI) (Vinh et al., 2010) between PMF factors and photochemical generation. The PMF residual is high for the 2-factor solution (13%, on average), and low for 3- to 10-factor solutions (less than 5%).

The normalized mutual information metric describes the correlation between PMF factors and generation. A value of 0 means no correlation, and a value of 1 indicates that generations are perfectly assigned to distinct factors. Because species can be assigned to multiple factors, we used the relative intensities of each generation in each factor as input to the NMI calculation. For instance, if PMF Factor 2 accounted for 66% of the total integrated intensity of first-generation product B1, 97% of the intensity of B2, and 12% of the intensity of B3, we assigned a value of 1.75 for first-generation products to Factor 2. The mutual information describes the probability that products of a particular generation are assigned to the same cluster. Mutual information must be normalized so that it can be compared between solutions with different numbers of factors or clusters. As the normalization factor, we used the arithmetic average of the generation and factor entropy, which is a quantity that describes the size and diversity of values in the two classification schemes (generation and PMF factor).

NMI values are provided in Table 1. For purposes of comparison, Table 1 also includes the NMI values calculated from hierarchical clustering analysis. HCA of the synthetic data set is described in section 3.2.1. Because there are only ten species in the synthetic data set, a solution with ten groups, each of which contains a single species, has no correlation between generation and groups, and the NMI is zero.

| Number of groups (PMF factors or HCA clusters) | PMF NMI | HCA NMI |
|---|---|---|
| 2 | 0.402 | 0.397 |
| 3 | 0.381 | 0.467 |
| 4 | 0.436 | 0.521 |
| 5 | 0.427 | 0.683 |
| 6 | 0.442 | 0.745 |

| | | |
|---|---|---|
| 7 | 0.761 | 0.835 |
| 8 | 0.733 | 0.799 |
| 9 | 0.679 | 0.756 |
| 10 | 0 | 0 |

**Table 1** Synthetic data. Normalized mutual information index quantifying the correlation between PMF factor or HCA cluster and photochemical generation.

Figure 3 shows the four-factor solution. The four PMF factors are able to reconstruct the total signal with excellent agreement, but they do not correspond to the four original generations of compounds (precursor plus three product generations). There is some relationship between early, middle, and late-generation species and the PMF factors (indicated by non-zero NMI values), but regardless of the selected rotational forcing, all PMF factors contain species from more than one generation. For instance, because both C1 and D2 are long-lived species, they are correlated over the time period of the experiment and so are assigned to the same factor. More importantly, many species are included in two or more PMF factors, despite being formed by only one pathway. Eight to ten factors (approximately the number of species in the dataset) are needed to separate generations, which is not a useful simplification of the data set (which is made up of only ten species).

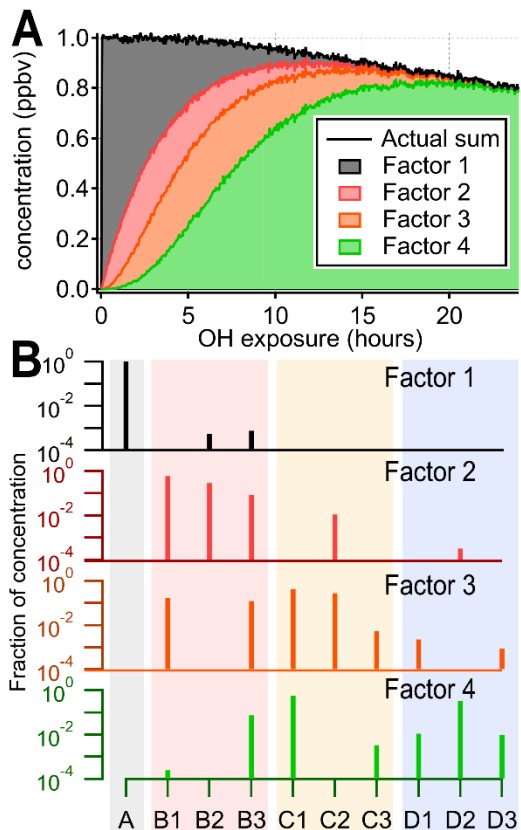

**Figure 3** Results from PMF analysis of the synthetic data set, showing the 4-factor solution. A. Total intensity of synthetic data compared to stacked time-series of PMF factors. B. Profiles of PMF factors, illustrating that factors do not correspond to individual generations. The shaded background corresponds to generation: precursor (black), first-generation (red), second-generation (yellow), third-generation (blue). The color of the mass spectra corresponds to panel A. Solutions with different numbers of factors are given in the supplemental information.

### 3.1.2 PMF of chamber data

Figure 4 illustrates positive matrix factorization of chamber data, including 463 individual calibrated product species from CIMS, optical, and AMS instruments; these exclude the precursor and overlapped species, and are corrected for background and dilution. A three-factor, four-factor, and six-factor solution are shown. Additional solutions are shown in Figure S6. In each of the solutions, a linear combination of PMF factors can

reconstruct the measured intensity with negligible residual (also called reconstruction error) (within 10 ppbC, or about 2%, for each solution, regardless of aging time). Each solution includes factors that peak in intensity at early, middle and late times. There are no factors that retain a consistent time-series or chemical profile between solutions with different numbers of factors, and in fact the time series do not have shapes consistent with chemical kinetics. Rather, each solution includes factors that peak in intensity at roughly regularly-spaced intervals, apportioning the time series into discrete pieces (Figure 4a). This suggests that the PMF factors are not chemically meaningful, even though the data are fit with low residual.

As in the PMF solution of the synthetic data set, most species appear in the profiles of more than one factor (Figure 4b). The time series of acetone (from calibrated $m/z$ 59 $C_3H_6OH^+$ measured by PTR-MS), a species with large signal and a long lifetime against OH, is shown in Figure 4c as an example. As oxidative aging progresses, acetone and other long-lived species, including butadione, acetic acid, and CO, are successively assigned to later-peaking factors, although mechanisms suggest that compounds such as butadione are formed in the first 1-2 generations of reaction (Bloss et al., 2005a; Jenkin et al., 2003; Li and Wang, 2014). Relatedly, two compounds that are formed in the same generation but exhibit different reactivity are not necessarily assigned to the same factor.

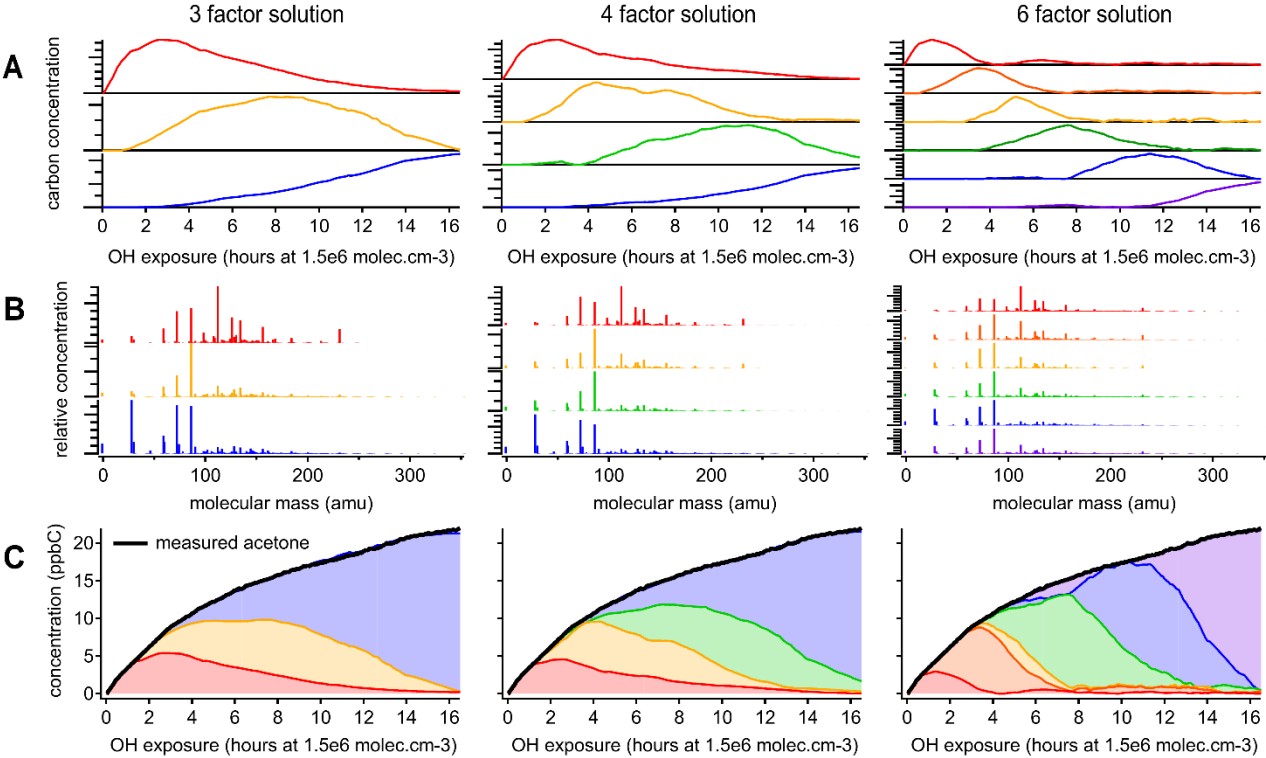

**Figure 4** Positive matrix factorization of chamber data, showing solutions with three, four, and six factors. A. Time series of PMF factors. B. Compositional profiles of factors, shown as combined mass spectra from all instruments with CO, CH2O, and CIMS measurements at their exact molecular masses and OA shown with a molecular mass of -1. C. Apportionment of the concentration of acetone (a long-lived oxidation product signal) across all factors. Within each column, the assigned color of each factor is consistent. As in the PMF analysis of the synthetic data set (Figure 3), factors do not correspond to generations, and long-lived species (such as acetone) are assigned to successively later-peaking factors over the course of the time series.

The chemical composition of each PMF profile can be summarized by calculating the average carbon oxidation state, and average number of carbon atoms per molecule in the factor (Figure 5). The contribution of each species to the average is weighted by its intensity in the factor profile. As the precursor species becomes more oxygenated and fragments to smaller product species, the average composition moves towards CO and $CO_2$, which are in the upper right corner of Figure 5 (Kroll et al., 2011). This trajectory is observed from early- to late-peaking PMF factors, as expected. Regardless of the number of factors chosen for the solution, the average chemical composition of each factor falls within the same range of oxidation state and molecular size. The various PMF factors appear to show the average composition of the mixture during early, moderate, and high OH

exposures. This is consistent with the time series of PMF factors, which appear at discrete intervals (Figure 4), and with the calculated average compositions of the mixture at specific time periods, which fall within the range of the PMF factors (Figure 5). In other words, solutions with a larger number of factors do not add new groups of species not represented by solutions with smaller number of factors, even though the PMF residuals are low.

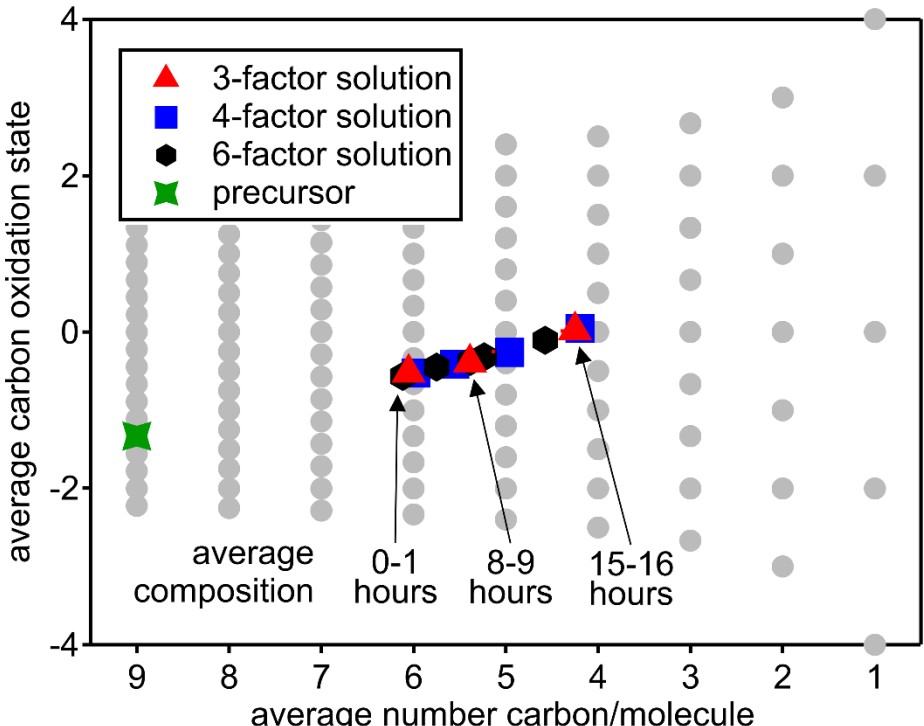

**Figure 5** Average carbon oxidation state and number of carbon atoms per molecule in each PMF factor from analysis of chamber data, for solutions with three, four, and six factors. Also noted is the average composition of the mixture during low (1 hour atmospheric equivalent aging), medium (8-9 hours), and high (15-16 hours) OH exposures. Factors cover a relatively small region in this chemical space, which is unaffected by the number of factors chosen for the solution.

We conclude that in chamber experiments such as the one considered here, the PMF factors generally
cannot be attributed to distinct chemical groups, oxidation generations, or chemical processes. Surrogate species derived from PMF factors do not have chemically realistic behavior or the same range of chemical properties as the original data set. The information about the system that can be determined from PMF factors is the average composition during specific time periods of the experiment. The researcher must subjectively choose the number of factors. These factors are not chemically robust and this should be considered when comparing PMF factors

between oxidation experiments or chemical systems. PMF is certainly well-suited for cases in which groups of
compounds have distinct and constant composition (Ulbrich et al., 2009), such as field measurements near fresh
emission sources, and/or when using instruments that classify mixtures into a small number of types (e.g., the
AMS). However, in a chamber oxidation experiment there are instead continuous, dynamic changes in
composition as a function of time. Species created in the same oxidation generation often do not have similar
time-series behavior, given differences in reactivity of different co-generated species. This could be a useful first-
level simplification of the data, but suggests that PMF factors derived from chamber experiments cannot be used
as surrogates for groups of reaction products within 3D models, because surrogate species should have chemical
behavior that emulates real species.

## 3.2 HCA

Hierarchical clustering can be used to identify major chemical groups in processed data. This could be
used to reduce the complexity of a dataset, by analyzing the chemical properties of the clusters rather than
individual species.

### 3.2.1 HCA of synthetic data

An example of the use of HCA to cluster chemical species within complex oxidation mixtures is shown
in Figure 6 using the synthetic dataset. Species D1 and D3, with very similar time-series behavior, are the two
most closely related compounds and are assigned to cluster D*. The next two most similar groups are species D2
and cluster D*, which are assigned to a new, higher-level cluster. Species are clustered together until all have been
grouped into a single cluster.

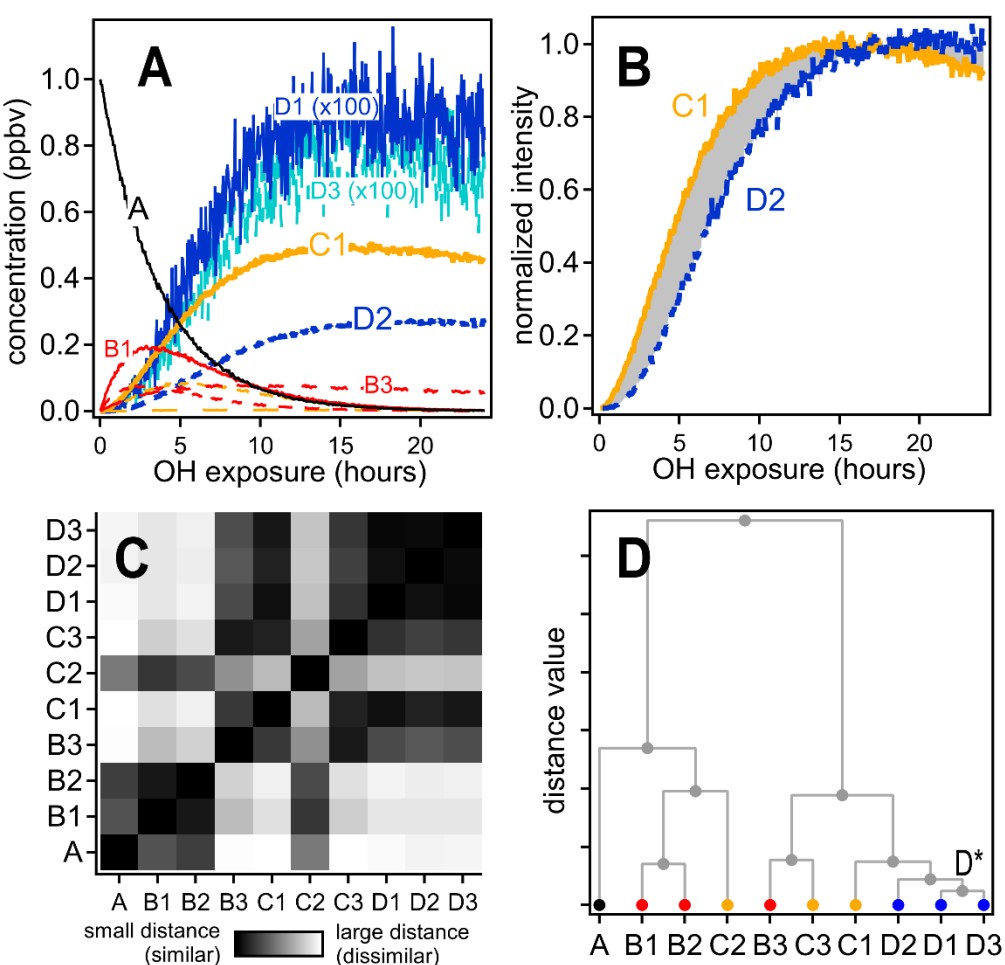

**Figure 6** Hierarchical clustering procedure applied to synthetic data. First, second, and third-generation species are shown in red, yellow, and blue, respectively, and the precursor is shown in black. A. Time series data. B. Time series of species C1 and D2 normalized between 0 and 1. The gray shaded area is integrated to give the distance between the two time series. C. Matrix showing the relative distance between each pair of species. D. Hierarchical cluster relationship; D1 and D3 are the most similar species, and so are the first to be clustered together (forming a new cluster D*)


In this example with simulated data, HCA generally clusters together compounds of similar generation, though not perfectly. HCA clusters together compounds that have similar time-series behavior, and time-series behavior is determined not only by generation, but also by formation and reaction rate constants. For example,

species B1, B2, and C2 all have fast formation and reaction rates, resulting in similar time-series. HCA groups these three species together. The algorithm suggests further that the first-generation products B1 and B2 are much more similar to one another, than they are to second-generation product C2.

The ability of HCA to separate compounds of different generations was quantified by the normalized mutual information (NMI). NMI values are provided in Table 1. For all solutions with more than 2 clusters (or factors), NMI values for HCA are higher than those of PMF, indicating that HCA more successfully sorts compounds by generation.

The results of HCA applied to synthetic data indicate several strengths and weaknesses of the HCA algorithm. Most importantly, the algorithm provides a clear way to visualize the behavior and relationships between all measurements in a dataset. The precursor compound can be included in the analysis, because data are normalized and the high intensity of the precursor does not skew the results. Compounds with similar kinetic properties are mostly grouped together, but some generational miscategorization still occurs. It may be difficult to use HCA to separate compounds which have different generation numbers but similar formation and reaction rates.

HCA can be used to simplify the dataset, by replacing clusters of compounds with surrogates. If the surrogate time-series behavior is determined by averaging the time-series of the individual members of the cluster, then the surrogate will have chemically realistic behavior. As noted previously, the researcher must subjectively choose the number of clusters.

**3.2.2 HCA of chamber data**

There are some significant differences between the synthetic data set, and real-world data sets collected from chamber experiments. Most importantly, the actual chamber experiment includes many more species (ten species in the synthetic system, compared to thousands of detected ion masses and hundreds of measured species in the chamber experiment). The real chamber data set includes many non-meaningful measurements whose time-series have no structure. Additionally, many species in the real-world data set have much more similar time-series behavior to one another than any two of the species in the synthetic system. Conversely, there are also distinct outliers in the real-world data set, whose time-series behavior does not resemble any other compound. HCA effectively separates meaningful from non-meaningful measurements, groups together very similar compounds, and highlights outliers.

A diagram showing the hierarchical distance between all species measured in the chamber study is shown in Figure 7. This data set includes measured, calibrated, background-subtracted species from all instruments, and

excludes overlaps. We use calibrated data here, but an advantage of this method is that it is insensitive to
calibration: data are normalized, and only relative behavior is important. In Figure 7a, individual species are
arrayed across the bottom, and their accumulation into clusters is denoted by gray lines linking species and
clusters. As with PMF, the user must choose the number of groups (factors or clusters) in the solution. Here we
have selected a maximum threshold relative distance that places the precursor, 1,2,4-trimethylbenzene, in a cluster
separate from all product species. The individual clusters that fall below this threshold are distinguished by color
in Figure 7a. The resulting groups include ten individual species that do not fall into a cluster (including the
precursor, 1,2,4-trimethylbenzene), and 9 clusters that incorporate at least two species. Figure 7b shows the time
series of a selection of these clusters (all time series are included in Figure S7). The cluster average was determined
by summing the individual species contributors to the cluster, weighted by parts-per-billion carbon.

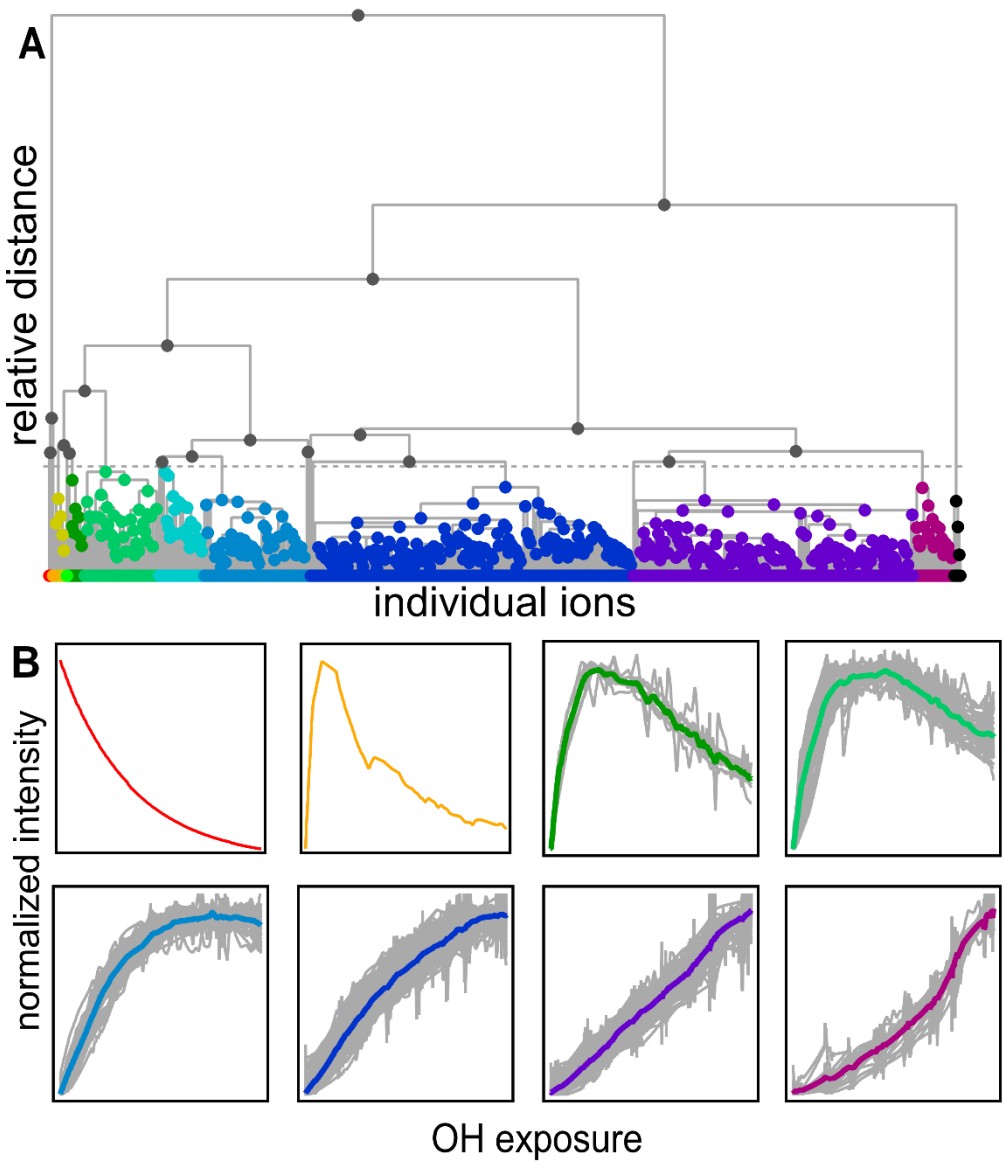

**Figure 7**A. Hierarchical cluster relationship of all measured species from the chamber experiment. Clusters are colored at a relative distance cut-off (gray dashed line) that separates 1,2,4-trimethylbenzene from all other products, with gray lines showing linkages between species and clusters. The individual clusters are distinguished by different colors. B. Time series of eight example clusters. The x-axis in each plot is OH exposure, and the y-axis is the normalized intensity. The cluster average

is shown by a thick colored line, and individual species contributors are shown as thinner gray lines. Colors correspond to those in Panel A.


The chemical properties of each cluster, described as average oxidation state and average number of carbon atoms per molecule, are shown in Figure 8. Clusters lie on a diagonal trajectory between the precursor and highly oxidized, small molecules (CO and $CO_2$), and clusters that peak earlier in time appear closer to the precursor. This indicates that species with similar time-series behavior have similar chemical properties.

Compared to the chemical properties of the PMF factors (Figure 5), the clusters lie along the same diagonal trajectory, but are substantially more varied in terms of average carbon number and oxidation state, and cover a wider range of chemical space. As the threshold for separating clusters is lowered, resulting in more clusters with fewer species per cluster, a wider range of chemical properties is observed (Figure S8). This is in contrast to PMF analysis, in which increasing numbers of factors does not increase the range of chemical properties (Figure 5). As

shown in Section 2.2.1., increasing the number of PMF factors provided the average composition of the mixture at more time points. HCA does not always separate generations perfectly (as can be seen in Table 1 and Figure 6d), but the generational mixing is not as severe as with PMF, and can be reduced by choosing a lower threshold for separating clusters.

The surrogate species derived from HCA clusters have chemically realistic behavior, and have a similar

range of chemical properties as the original data set. As with PMF, the choice of the number of clusters is subjective. In addition to defining surrogate species, HCA can be used to visualize the range of behavior and degree of similarity between all compounds in a data set. The clustering algorithm is thus a viable approach for describing a continuum of kinetic behavior and chemical properties.

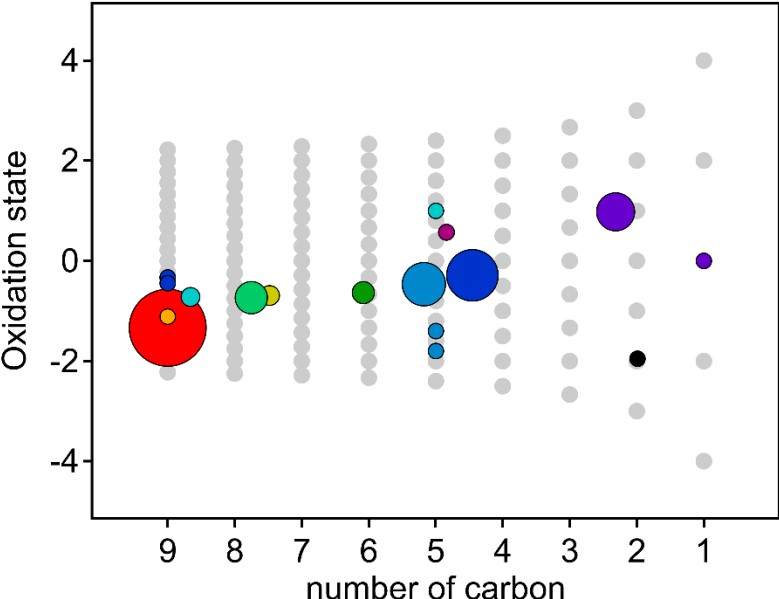

**Figure 8** Average oxidation state and number of carbon atoms per molecule for each cluster determined from HCA of chamber data. The individual clusters are distinguished by color, and the color scheme is the same as in Figure 7. The contribution of each species to the cluster average is weighted by parts-per-billion-carbon (averaged over the entire experiment). Marker area is proportional to the averaged concentration (parts-per-billion carbon) of all species in the cluster, with the marker size of the precursor (red) decreased by a factor of 2 for legibility. Clusters cover a substantially wider area of chemical space than PMF factors (Figure 5).

## 3.3 GKP

### 3.3.1 GKP fit to synthetic data

The gamma kinetics parameterization (GKP, Eq. 3) provides a method for determining kinetic and mechanistic information from chamber experiments. The parameterization returns an effective rate constant $k$ and generation number $m$. To investigate the extent to which fitting kinetic data to Eq. 3 yields reasonable values for rate constants ($k$) and generation number ($m$), we first apply the parameterization to the synthetic data set described in section 2.1.2, which has known rates and generation numbers. Figure 9a shows the time series of synthetic data and the parameterized best-fit, using integrated signal as described in Section S1. The parameterization can reproduce a range of kinetic behavior, even in situations where the formation and loss rate constants $k_m$ are very different (for

which the assumption of uniform reactivities is poor). Figure 9b shows the fitted generation compared to the actual

generation. The actual generation numbers are correctly returned in all cases (with errors within 12%). Figure 9c

shows the parameterized $k$ compared to actual pathway-average $k_m$ rate constants in the pathway. The effective

rate constant $k$ cannot be calculated directly from the actual $k_m$ in the system, but is rather a best-fit value in the

approximation of equal $k_m$. The returned values of $k$ are in the same range as the actual $k_m$, and are larger for

pathways that generally involve faster rate constants. The average rate constant in a particular pathway and the

fitted effective rate constant $k$ are similar, except when the pathway includes a very slow step. In this case the

fitted value of $k$ is closer to that of the rate-limiting step (Figure 9c). We conclude that the fit parameters $m$ and $k$

are reasonable, physically meaningful values that provide information on the kinetics of the system.

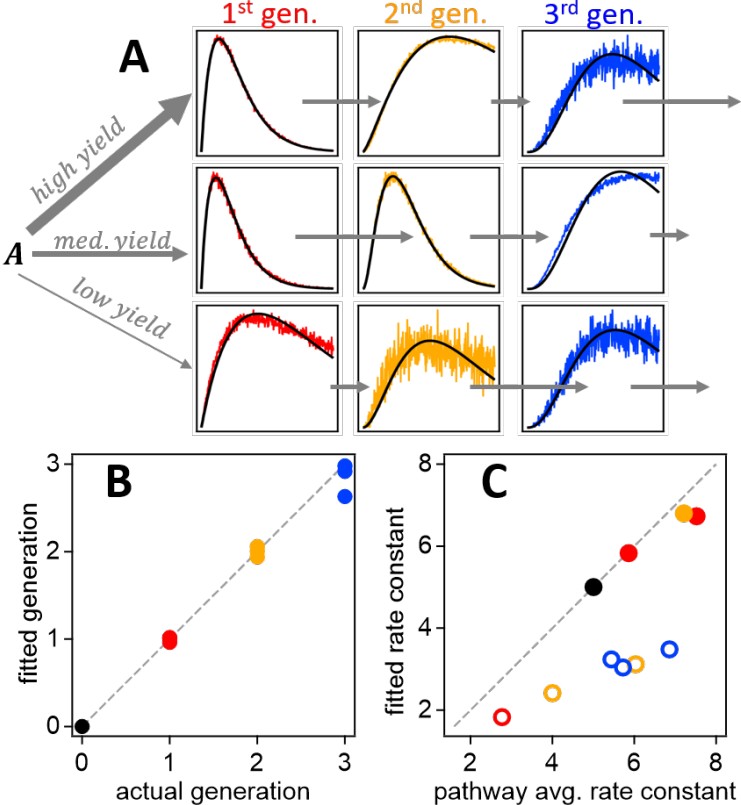

**Figure 9** Best fit of the gamma kinetic parameterization to synthetic data (GKP, Eq.3). A. Time series of synthetic data (colored

lines) and best-fit (black lines). First-generation species are shown in red, second-generation in yellow, and third-generation

in blue. The relative rate constants are indicated by short arrows (slow rate constant) or long arrows (fast rate constant). B.

Fitted generation compared to actual generation. The colors correspond to the generations shown in panel A. C. Effective rate

constant compared to the average of the rate constants in the pathway that produces each particular species. Pathways that include slow steps are shown with open circles.

### 3.3.2 GKP fit to chamber data

The GKP was applied to the chamber data, with the time dependence of all measured compounds fit to Eq. 3. More than 95% of measured compounds are fit with a correlation coefficient $R^2$ of 0.9 or higher, meaning the function generally describes well the kinetic behavior of species measured in oxidation systems. Examples of fitted chamber measurements are shown in Figure 10. In some cases, non-integer values of $m$ are returned, which may occur for several reasons.

First, noise can contribute to uncertainty in $m$. At low generations ($m$=1-2), the standard deviation of the fit is about 0.1, and at high generations ($m{\geq}3$) is somewhat higher, with standard deviation up to 0.8 (Figure S9). Especially for measurements with low signal-to-noise ratios and limited data near the beginning of the experiment, $m$ may not be fit with high precision. For example, the fits using $m$=3 and $m$=5 to $C_5H_6O_6$ (Figure 10i) are not significantly worse than $m$=4.

Second, the generation number can be distorted if the compound is produced by or reacts significantly via channels other than OH reaction (e.g., by ozone reaction, $NO_3$ radical reaction, or photolysis), in which case the assumption of linear, first-order kinetics with respect to OH exposure is not necessarily applicable. For example, $C_6H_8O_2$ (Figure 10c) may correspond to 3,4-dimethyl-2(5H)-furanone (Bloss et al., 2005b), which reacts with $O_3$ under experimental conditions at a comparable rate to OH, or dimethylbutanedial (Li and Wang, 2014), which has a high photolysis rate. In Figure 10b and c, the curves are also distorted due to repeat injections of HONO, which abruptly changes the NO concentration in the experiment and clearly affects the reaction of these compounds. Any of these processes can distort the shape of the curve, making it more difficult to fit $m$ correctly. Because $m$ is related to the slow (rate-limiting) steps in a mechanism, specifically OH additions, it is not affected by faster radical chemistry such as autooxidation and intramolecular arrangements.

Finally, if the compound is produced by more than one pathway with a differing number of reaction steps, such as butadione (Figure 10d), the resulting generation parameter is non-integer. This is also demonstrated using a synthetic system in Figure S10.

In addition, if physical (non-chemical) processes have a major influence on species concentrations, and occur on the same time scale as the chemical reaction, they may impact the fitted kinetic parameters. In particular, delays caused by strong interactions of gas-phase compounds with surfaces (chamber walls or instrument inlets) can shift the fitted $m$ to higher values and the fitted $k$ towards the time constant of the surface interaction. As noted

above, the timescales of surface equilibration processes in the present experiments are <15s, much shorter than the timescales of the chemical changes observed. Thus such processes are unlikely to affect the analysis of the present chamber results, but could introduce substantial errors if they occur over longer timescales, or are competing against much more rapid chemical transformations. GKP analysis is therefore only valid when the equilibration times of such processes are short compared to the timescales of the chemical processes being studied.

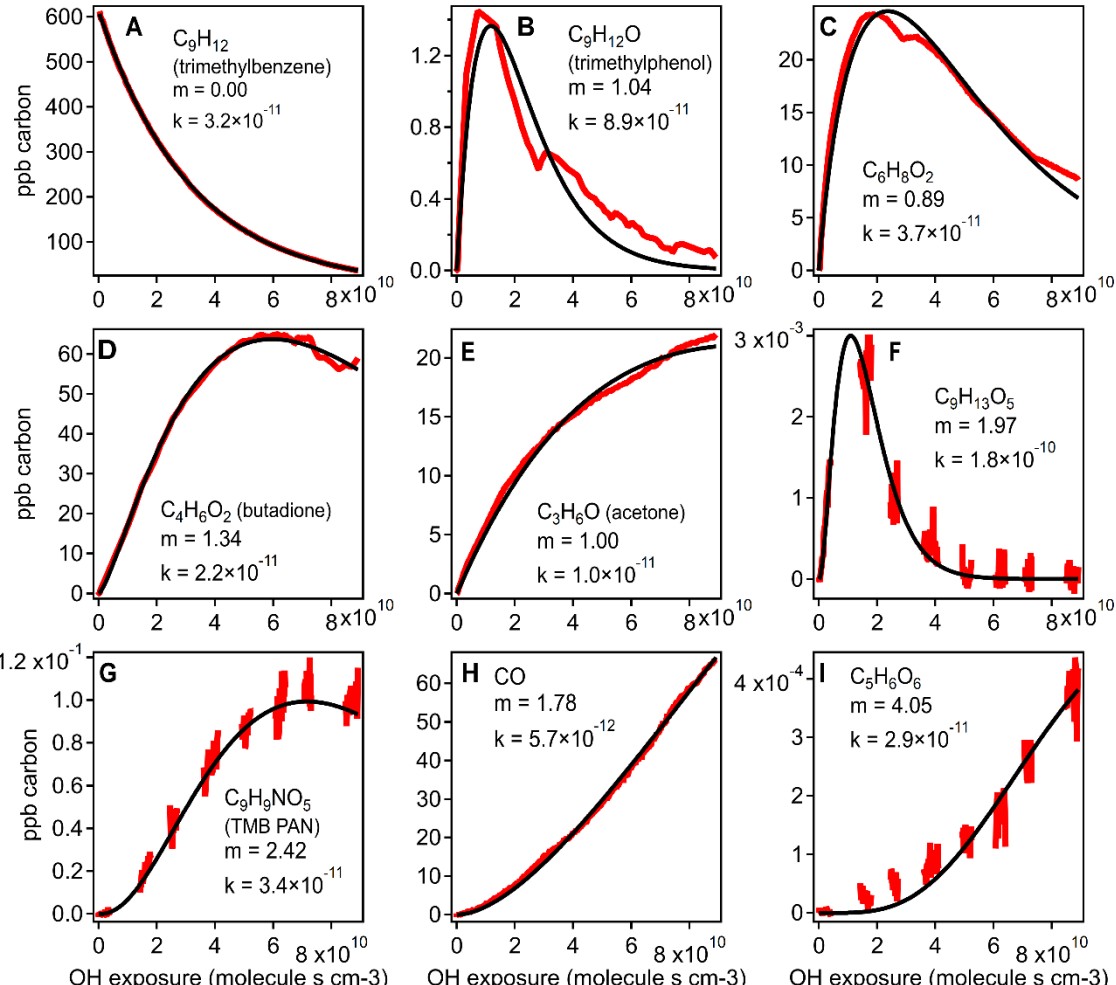

**Figure 10** Measured species from chamber experiment (red) and GKP best fit (black). Data in panels A, C and E are from Vocus-2R-PTR; in panels B and D from PTR3-$H_3O^+$, in panels F, G, and I from I⁻ CIMS, and in panel H from TILDAS. The data gaps in panels F, G, and I arise from the I⁻CIMS instrument measuring particle-phase composition, measurements that are not considered in this work.

The fitted values of $k$ and $m$ for all species are shown in Figure 11. The returned $k$ fall within one order of magnitude of the OH rate constant of the precursor species ($k_{TMB} = 3.2 \times 10^{-11}$ cm$^{-3}$ molecule$^{-1}$ s$^{-1}$). Most *"m"* are between 1 and 2, meaning most measured compounds are produced after one or two reaction steps (assuming OH is the dominant oxidant). When the data are restricted to fast-reacting compounds, major modes at integer values of $m$ are observed (black bars in Figure 11). However, when all compounds are considered, major modes at integer values are not observed, which suggests that many compounds are formed by more than one pathway, and/or have significant reactions with O$_3$ or another oxidant. The generation numbers of compounds with $m>=4$ are less certain due to data gaps, limited experimental duration, and low signal-to-noise ratio in the fits. Higher-generation ($m>2$) compounds are uniformly the fast-reacting (high $k$) species. Conversely, no species are observed with high $m$ ($>2$) and low $k$. This area of the diagram corresponds to slow-forming, slow-reacting species that are created after multiple OH additions; such species are unlikely to be formed at observable concentrations within the timeframe of the experiment. Were the experiment to be run at higher OH exposures, it is possible that these species would be observed as well.

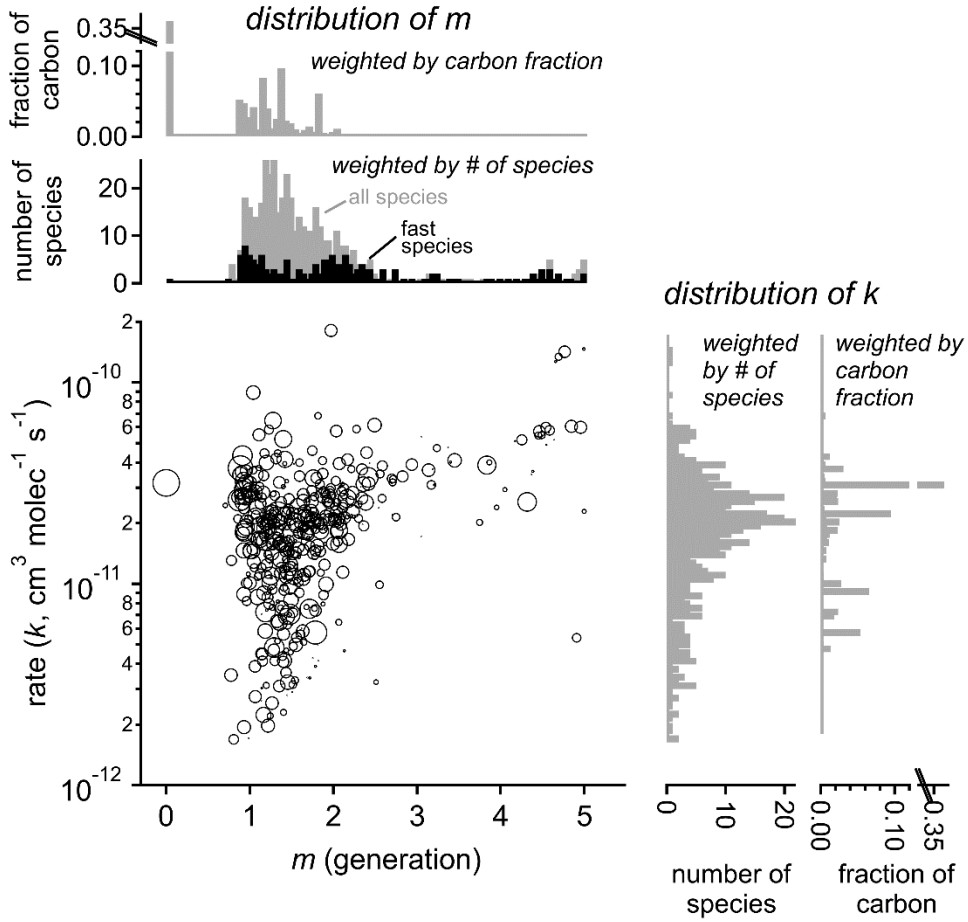

**Figure 11** Parameterized rate constant and generation number for 463 species detected during the chamber experiment OH-initiated oxidation of trimethylbenzene. Marker area corresponds to log(ppb carbon) of detected species, averaged over the
duration of the experiment. "Fast-reacting" species, defined as having an effective rate constant at least 75% that of the precursor, are highlighted as black bars in the histogram of $m$. These tend to center on integer values of generation number.

The kinetic parameters derived from fitting the gamma distribution are correlated with individual species' chemical composition. Figure 12 shows that species that involve the fastest reactions (high values of effective rate constant, $k$) and earliest formation (lowest values of $m$) tend to be large and relatively unoxidized, with oxidation
states similar to that of the 1,2,4-trimethylbenzene precursor. Species that form or react slowly (low values of $k$) or that form in later generations (higher values of $m$) tend to be smaller and more oxidized.

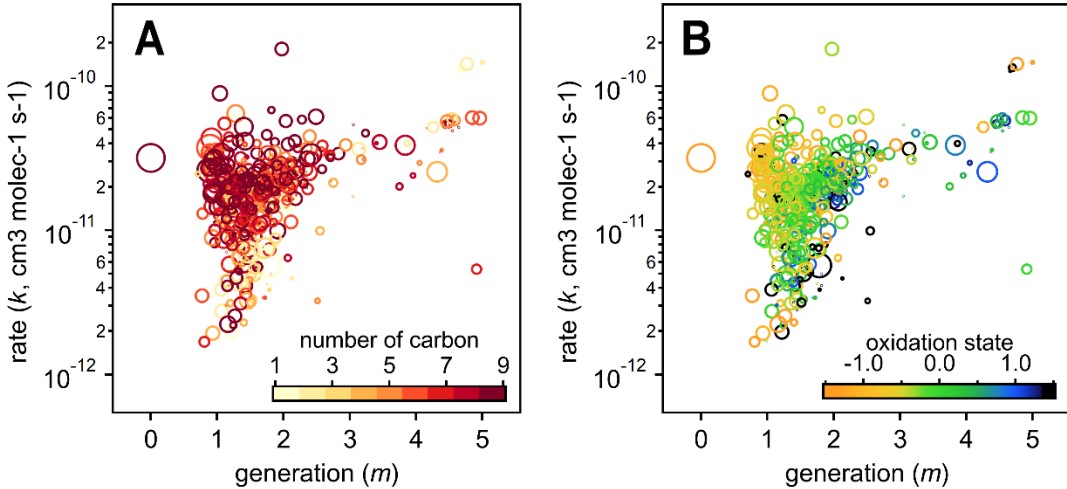

**Figure 12** Relationships of kinetic parameters (from the GKP of chamber data) with key chemical properties of reactive species. A. Generation ($m$) and rate ($k$) values of 1,2,4-trimethylbenzene precursor and products, colored by number of carbon atoms. B. Same as A, but $k$ and $m$ colored by carbon oxidation state. Marker area corresponds to log(ppb carbon). The early generation and fast-reacting products tend to have higher numbers of carbon atoms and are less oxidized, while later generation and slow-reacting products tend to be smaller and more oxidized.

### 3.3.3 Clustering of GKP results

The GKP can be used not only to describe individual species, but also to group compounds and reduce the complexity of the system. If compounds are grouped by similar $k$ and $m$, compounds in the group will have similar chemical composition and similar kinetic behavior, and the chemical and kinetic properties of the groups will include a range of variability similar to the individual species. Here we test three methods of using GKP to group compounds: (1) fitting the GKP to time-series of HCA-derived clusters; (2) using HCA to cluster compounds based on their GKP-derived time-series (based on fitted values $k$ and $m$); and (3) using fixed bins to group compounds based on $k$ and $m$. Groups derived from PMF analysis cannot be fit with the GKP because the factor time series are not consistent with chemical kinetics.

Results from each approach, showing both kinetic characteristics ($k$ and $m$) and chemical properties (oxidation state and carbon number) of each group, are given in Figure 13, which includes an overview and comparison of grouped species derived from PMF (Figure 13a), HCA (Figure 13b), and GKP (Figure 13c and d). Figure 13b shows results from applying the GKP to HCA data. For each of the nine HCA clusters (described in Section 3.2.2), the GKP was fit to the cluster's average time series, determined from a carbon-weighted average

of the time series of all individual species in the cluster. This provided values of $k$ and $m$ for each cluster. (For the ten species that did not fit into any cluster, the $k$ and $m$ of these were determined as well). Figure 13c shows the reversed approach, the application of HCA to GKP results. Here, the time-series of each individual species was fit with GKP, and the distances between the time-series of the best fits were determined and used as input into the HCA algorithm. The $k$ and $m$ of the resulting cluster were calculated by averaging the $k$ and $m$ of the individual compounds in the cluster, weighted by parts-per-billion carbon. A potential advantage of this approach is that the GKP fitting reduces the noise of the signals used in HCA analysis, possibly allowing for more precise determinations of clusters. Finally, shown in Figure 13d are results from an alternate approach for grouping compounds by GKP parameters ($k$ and $m$), binning all the species by their values of $k$ and $m$. This is analogous to the 2D-volatility basis set developed by Donahue et al. (2011, 2012), which bins species based on saturation mass concentration and O:C ratio.

Surrogate species defined by GKP have by definition kinetically realistic behavior. The resulting groups of compounds have a range of chemical properties similar to that of the original data set, regardless of whether they are grouped using HCA or grouped by similar $k$ and $m$. The method of grouping is subjective, as is the choice of number of clusters (if HCA is used) or the number of bins (if compounds are grouped by similar $k$ and $m$). A particular strength of GKP is the resulting kinetic characterization of each compound. The effective rate constant and generation number provide new information that can be used to assess proposed mechanisms or to guide the reactive behavior of surrogate species in a model.

### 3.4 Comparison of PMF, HCA, and GKP

A comparison of compound groups derived from PMF, HCA, and GKP is shown in Figure 13. This figure shows the chemical properties (oxidation state vs. number of carbon atoms), time series, mass spectra, and kinetic properties ($k$ vs $m$) of the compound groups. For each technique, solutions with different numbers of groups are possible. Figure 13 shows the solution discussed most extensively in the text: the six-factor solution for PMF; the HCA solution with nine major clusters; and the two GKP solutions discussed in section 3.3.3, which have seven major clusters and twenty-five bins, respectively. For clarity, the time series and mass spectra for only six groups derived from HCA and GKP are shown. These six groups contain cumulatively about 80% of the total product carbon in the system.

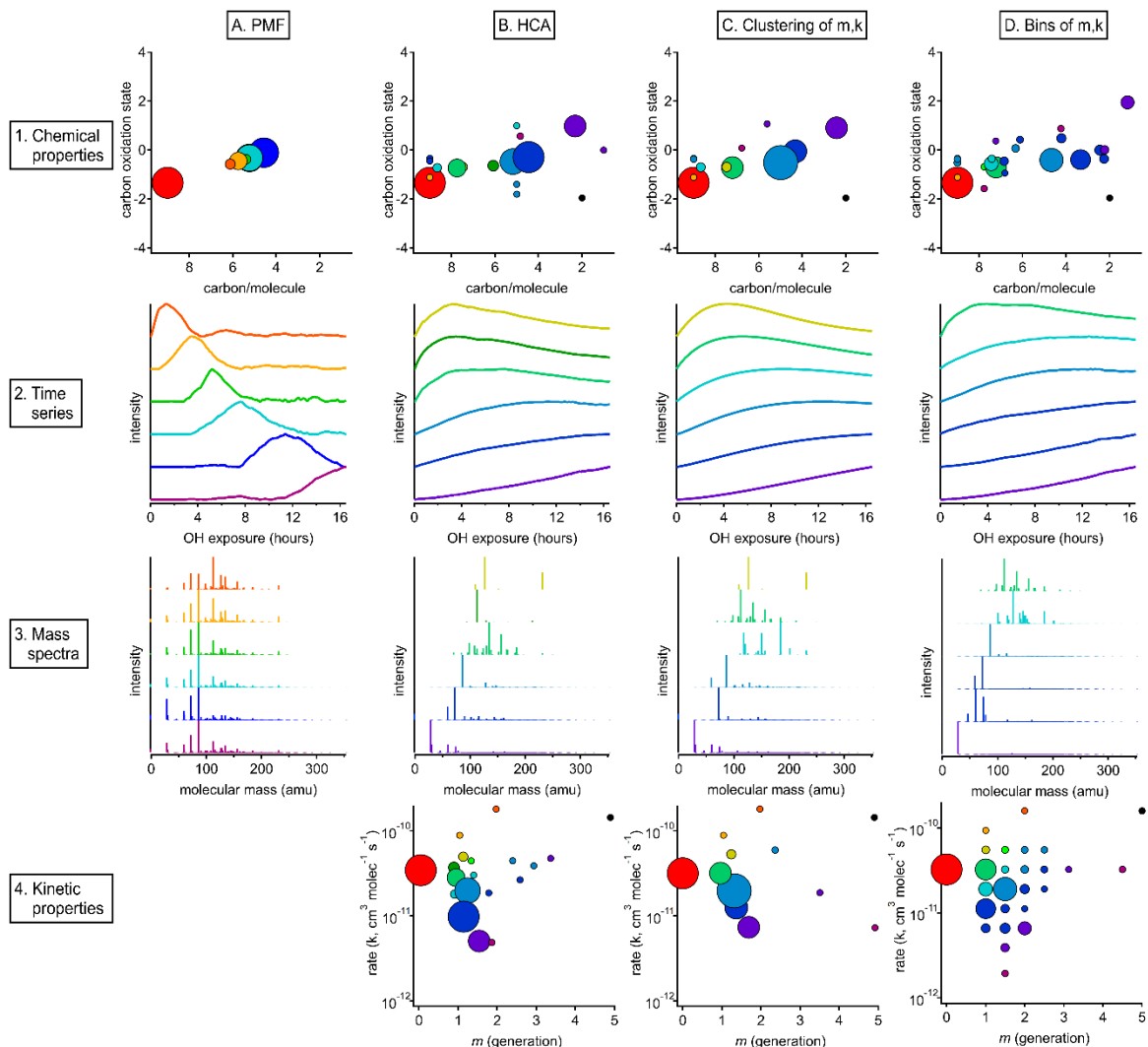

**Figure 13** Overall comparison of groups derived from PMF, HCA, and GKP of chamber data. The columns show, from left to right, the results of A. PMF, B. HCA, C. GKP best-fits grouped using HCA, and D. measurements grouped by GKP fit parameters. The rows show, from top to bottom, 1. the average carbon oxidation state and number of carbon atoms per molecule for each group, 2. the time series of the six groups containing the most carbon, 3. the mass spectra of those six groups, and 4. the rate constant and generation number of each group. Within each column, each chemical group is assigned a specific color. This color scheme is the same for each plot within a column. The marker area is proportional to the averaged concentration (ppb carbon) of all species in the group, with the marker size of the precursor (red) decreased by a factor of 2 for legibility. The marker area scheme is consistent across all plots. PMF factors do not have kinetically realistic time series, therefore there is no plot A4.

720

725

In all cases, the majority of the carbon can be represented by a manageable number of groups. The relationship between oxidation state and number of carbon per molecule is similar, regardless of the grouping technique. The PMF factors have a smaller range of chemical properties than chemical groupings derived from HCA or GKP. The range of chemical properties is similar for HCA and GKP. The time-series of PMF factors are clearly different from those of HCA- and GKP-derived groups, and have non-kinetically-realistic shapes with sharp maxima.

The PMF factors each contain many more compounds than the groups derived from HCA or GKP. Many of the same compounds are consistently grouped together by HCA and GKP, regardless of whether HCA, HCA of GKP, or binning of GKP is used. Additionally, the range of kinetic properties, and the locations of major compound groups in kinetic space, are similar between the HCA and GKP approaches. This reproducibility suggests that these are chemically meaningful compound groupings. Some groups derived from HCA or GKP contain only a single species. These could be chemically important compounds whose unique behavior should be considered when modeling the system; conversely, they could be measurement outliers which should be discarded. The interpretation of these species is subjective.

Regardless, the combination of fitting using the GKP and grouping based on kinetic behavior may provide a viable approach for greatly simplifying the time-dependent behavior of complex mixtures of reaction products in a laboratory oxidation system.

## 4 Conclusions

Hundreds to thousands of individual chemical species can be produced in a typical organic photooxidation chamber experiment. This chemical complexity presents a number of analytical challenges, including organizing and processing large mass spectrometric data sets, identifying major groups of compounds, providing kinetic and mechanistic information, and simplifying the chemistry in a way that can be implemented in large-scale regional and global models.

In this paper, we evaluated three methods to simplify a description of atmospheric chemistry in chamber studies. The methods explored include positive matrix factorization (PMF), which represents data as a linear sum of factors, hierarchical clustering analysis (HCA), which describes similarity of species in terms of their time-series behavior, and the gamma kinetics parameterization (GKP), which characterizes species in terms of effective rate constant and generation. All three approaches require a subjective choice of the number of compound groups.

Because PMF is so widely used in atmospheric chemistry to characterize organic aerosol and for source apportionment in field studies, it is important to understand how oxidation systems are described by PMF. We found that PMF analysis of the chamber experiment described here did not sort species into clear generations, since different species formed in a single generation can exhibit highly variable reactivities. Oxidized factors appearing in PMF analysis of chamber studies, and in ambient air, may be able to reproduce observations as a

linear sum of a "fresh" factor and a "highly aged" factor with low residual, but these factors do not necessarily represent distinct chemical groups. This is because PMF assumes constant factor composition, which is useful when distinguishing fresh emission sources, but does not apply to evolving oxidation systems.

Hierarchical clustering, which also does not depend on calibration, can be used to quickly identify major groups of ions and patterns of behavior. The derived clusters maintain more chemical information (including

average oxidation state and molecular size) than do PMF factors. HCA is therefore useful to identify chemically meaningful ions in mass spectrometry data, and to group compounds into a smaller number of groups with consistent chemical characteristics.

A continuum of kinetic behavior is observed and can be described using the gamma kinetics parameterization of individual species (or clusters of species). The parameterization is derived from first-order kinetics and thus

provides a physically meaningful fit to the kinetics of the species. The two returned parameters, effective rate constant and generation number, correlate with oxidation state and molecular size. The parameterization provides a way to derive mechanistic information from an oxidation system, in addition to describing chemical composition.

Future directions of this work include evaluation of mechanisms, mechanism development, and applications

to lumping schemes in models. The current analysis is based on two systems, a synthetic system and a chamber experiment, and more work is needed to see how these analysis approaches perform with other systems. The gamma kinetics parameterization can be used to support complex chemical mechanisms, by determining whether the experimentally determined generation and rate constants are consistent with a proposed pathway or mechanism. Further, with well-calibrated, high-quality laboratory data, it may be possible to derive yields,

formation rate constants, and reaction rate constants separately, which would be invaluable in model and mechanism development. Finally, HCA-derived clusters, or groups of compounds with similar effective rate constant and generation, could be used as surrogates or "lumps" in aerosol or air quality models, as an experimentally supported way of simplifying a complex system.

**Acknowledgements**

Author contributions: ARK, MRC, AZ, JEK, MB, KN, CL, JCR, and JRR collected and analysed data. ARK implemented PMF and HCA algorithms, developed the GKP analysis, and wrote the manuscript. FNK and JHK provided project guidance. All authors were involved in helpful discussion and contributed to the manuscript. This work was supported by NSF grant AGS-1638672. We additionally acknowledge the Harvard Global Institute for funding. ARK acknowledges support from the Dreyfus Postdoctoral program. MB acknowledges support from

the Austrian science fund (FWF), Erwin-Schrödinger-Stipendium, grant J-3900.

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
