# Peer review of "Dimensionality-reduction techniques for complex mass spectrometric datasets: application to laboratory atmospheric organic oxidation experiments"

_Atmospheric Chemistry and Physics, 2019_

## Referee Comment (RC1) · Anonymous Referee #1 · 23 Jul 2019

Koss et al. present three statistical/mathematical approaches to reducing the time series of multiple species observed or simulated in a laboratory chamber into chemically meaningful groups or clusters. The authors conclude that the PMF (positive matrix factorization) technique does not perform nearly as well as the HCA (hierarchical clustering) or GKP (gamma kinetics) techniques in binning compounds (observed or simulated) into proper generational groups that share common chemical ages. The manuscript is concise, well written. Figures are illustrative, support the reported conclusions. This review points to a few areas of ambiguity that should be addressed/clarified.

[Figure]

The work is appropriate for publication in ACP after these minor revisions.

PMF does a poor job compared to HCA at assigning members into chemically meaningful groups. This result is a big deal given how widely PMF is used. More discussion and/or tests are needed to explain exactly why the PMF does not perform as well, that is, can PMF be modified so that it performs as well as HCA? Mathematically, matrix factorization and hierarchical clustering are similar. They certainly have the same intended goals. One difference is that with HCA each time series is normalized such that all compounds are more or less given equal weight, whereas PMF is biased towards those with higher signal to noise. Though it is not standard operation, it would be useful to re-run PMF on the $\sim$400 TMB products but after normalizing each member such that they are given equal weight. Basically, feed the same input to HCA and to PMF. Let's compare apples to apples. Doing so will rule out differing inputs as the reason for the different performance. Also, constrain PMF such that each member can belong to only one factor.

It would be helpful to make mirroring figures so that comparing the performance of each of the three techniques is easier. For instance, make figure 7 for PMF look like figure 9 for HCA. Include a figure like figure 6b (mass spectrum of each factor) but for HCA results as part of figure 8. Same with GKP. Is the reason that PMF factors do not accurately represent chemical age groups is because each factor contains compounds with a wide range of amu (as shown in figure 6)? Is this not the case for HCA (please show in figure 8). And also for GKP.

The way that HCA is described on page 12, it reads as if the technique also solves for the final number of clusters needed to explain the variance of all input members. But it turn out (page 23 line 465) that this final cluster number is "chosen" by the user. How was this final number objectively determined? There are eight clusters (figure 8), but that can easily be reduced further (6, 7 and 8 look pretty similar, as do 3 and 4; and conversely, each of the those eight can be split even further). PMF at most has 6 factors. GKP has 9. How are the final group/factor/cluster number chosen for PMF and

GKP? Perhaps the authors should choose the same number for the three techniques. This again, I think will present a fairer comparison of the three techniques.

There are sections dedicated to PMF (section 3.1) conducted on both simulated and chamber data. Same with GKP (section 3.3). Why not HCA (section 3.2)?

Minor Figure 1 bottom. legends needed.

page 7, line 167: "Teflon" itself is PTFE and is trademarked and manufactured by a company called Chemours. Is the tubing PFA or PTFE? Manufactured by Chemours? If not, it is not Teflon.

Figure 2. Panel D. Not a great figure to highlight in main manuscript. Account for oscillation before including as main figure. I understand it is not included, but most people only look at figure and not read caption, will come away with wrong impression. Panel B not informative. This figure perhaps is introduced prematurely since hierarchical clustering since that section is far below.

Page 12 line 290, Need hyphen in citation

Figure 8, 9 and 14 share the same color scheme. It would be nice to have a common legend and/or color-bar shown in each of these figures to remind the reader that these colors represent generation determined by HCA.

Please make clear in each of the figure caption in the SI and main manuscript whether it is presenting simulated data or measured data.

---

## Referee Comment (RC2) · Anonymous Referee #2 · 28 Sep 2019

The authors compare three different techniques for dimensionality reduction in mass spectrometry time series data sets, positive matrix factorization, hierarchical clustering, and gamma kinetics parameterization. They evaluate the behaviour of the three techniques on two data sets and conclude that PMF is not competitive compared to the two others.

Overall the paper gives a good overview of the work, but requires some revision before being published.

[Figure]

For the clustering, the authors chose agglomerative clustering using average linkage. However, they don't provide all necessary information to reproduce the experiments or motivate their choices. Euclidean distance was used, why was this distance measure chosen? I can see it makes sense in some ways, however there are different metrics specific for time series, most notably dynamic time warping (DTW), which might make sense in this case. DTW compensates for shifts in the time series, so for particular use cases, this could make sense. Otherwise, there are more approaches to achieve a clustering, why chose this one? Agglomerative tends to be computationally faster than divisive, but this shouldn't be a problem with data sets this size. Otherwise what about density-based clustering or really simple approaches such as K-Means?

Similarly, there exist a wide range of algorithms for PMF, which one was actually used here? And why this one? The authors give the library, but some details on the method would be necessary.

On page 11, the authors give a formula for the quality of fit parameter Q, but half of the variables in the formula are not defined anywhere so the formula does not really make sense.

Also, when looking at algorithms such as PMF or clustering, it would be interesting to calculate performance measures and give them to get a feeling how well the clustering or factorization works. This could be simple reconstruction error or normalized mutual information (if there is a ground truth).

Another issue is repetition of experiments. While the agglomerative clustering should be mostly stable, PMF usually is not when using big enough data sets. So a single run would not be a reliable representation and multiple runs would be necessary. Additionally, this paper seems to base its results on two data sets, which cannot give any reliable or statistically sound performance representation for these approaches. Anything below at least 5 data sets won't give you the proof you need for what you state in the conclusion. Either rerun the experiments a lot more times or restate in the conclusion that this gives an indication, but to proof it, many more experiments would be needed.

I would recommend improving the presentation, particularly the figures. It is not always straight forward to understand what is shown. For example Figure 14 on its own does not explain the meaning of the colours or the size of the dots. Also the labeling of part A, B, and C is not very standard and sometimes hard to understand.

One specific question I have on Figure 2. I don't see how the cluster of C happens, looking at the data in C, this does not seem to be a cluster, the highest gray line is far away from all others and looks far too much as an outlier compared to the rest of the cluster. How does this compare to the rest of the data? Is everything else just even further away?

---

## Author Comment (AC1) · 22 Nov 2019

**ACP-2019-467**

**Response to reviewers**

We thank both reviewers for their positive comments, as well as their critical remarks, which have helped make the paper more organized, clearly expressed, and scientifically robust. As a whole, both reviewers suggested a more direct comparison of the three techniques, as well as a more direct statement of our evaluation criteria. To address this, we have made the following edits to the introduction:

Page 2, line 89 now reads,

"*Lumping schemes could be improved by using laboratory data to define important groups of compounds, and assign experimentally-derived chemical and kinetic properties to each group **to act as a surrogate species***."

The last paragraph of the introduction now reads,

"*The three methods (PMF, HCA, and GKP) have different mathematics but the same goals: to identify groups of compounds, and replace each group with a chemically meaningful surrogate. The three methods are evaluated in the following criteria: whether the resulting surrogates have chemically realistic behavior; whether the surrogates have the same range of chemical properties as the original data set; which subjective choices the researcher needs to make when implementing the method; and what other new information about the system can be learned. We additionally discuss the extent to which different methods agree in their identification of major groups of compounds. The output of these dimensionality-reduction techniques can be used to quickly analyze and interpret chamber experiments, and could be used to reduce the complexity of chemical mechanisms included in models.*"

Below we respond directly to specific reviewer comments.

**Anonymous Referee #1**

Koss et al. present three statistical/mathematical approaches to reducing the time series of multiple species observed or simulated in a laboratory chamber into chemically meaningful groups or clusters. The authors conclude that the PMF (positive matrix factorization) technique does not perform nearly as well as the HCA (hierarchical clustering) or GKP (gamma kinetics) techniques in binning compounds (observed or simulated) into proper generational groups that share common chemical ages. The manuscript is concise, well written. Figures are illustrative, support the reported conclusions. This review points to a few areas of ambiguity that should be addressed/clarified.

The work is appropriate for publication in ACP after these minor revisions.

PMF does a poor job compared to HCA at assigning members into chemically meaningful groups. This result is a big deal given how widely PMF is used. More discussion and/or tests are needed to explain exactly why the PMF does not perform as well, that is, can PMF be modified so that it performs as well as HCA? Mathematically, matrix factorization and hierarchical

clustering are similar. They certainly have the same intended goals. One difference is that with HCA each time series is normalized such that all compounds are more or less given equal weight, whereas PMF is biased towards those with higher signal to noise. Though it is not standard operation, it would be useful to re-run PMF on the~400 TMB products but after normalizing each member such that they are given equal weight. Basically, feed the same input to HCA and to PMF. Let's compare apples to apples. Doing so will rule out differing inputs as the reason for the different performance. Also, constrain PMF such that each member can belong to only one factor.

We address three points mentioned by the reviewer in this comment: (1) usefulness of PMF, (2) mathematics of PMF vs HCA, (3) applying PMF to normalized data.

(1) PMF works extremely well for many applications. For example, PMF has been used for many years to interpret data from aerosol mass spectrometry, and we do not dispute that these interpretations are useful and scientifically meaningful. Here, however, we show that PMF is not always the best choice of analysis tool for one specific, but important, application: chamber oxidation experiments.

We do not conclude that PMF is broadly inferior to HCA. Our aim is that this work will help researchers to choose an analysis technique, and interpret the results, depending on the features and goals of the experiment in question. To clarify this, we have edited in section 2.2.1 (description of PMF):

*"**PMF analysis of ambient air measurements has in many situations been shown to be robust and meaningful, and has contributed greatly to our understanding of atmospheric and aerosol chemistry.** PMF is frequently used for source apportionment and characterization of organic aerosol in field studies, for example, to sort aerosol as more- or less-oxidized, or from a specific source such as biomass burning (Zhang et al., 2011). PMF is also frequently applied to VOC measurements in field studies. In this application, each factor indicates a particular VOC class (which can be associated with a specific source) and its magnitude, which is a powerful tool to support regulation.*
*        **Some aspects of atmospheric chemistry can complicate PMF analysis.** Oxidation chemistry during transport from the source to the measurement location can change the chemical composition…"*

In the discussion of PMF, Section 3.1.2:

*"We conclude that **in chamber experiments such as the one considered here**, the PMF factors generally cannot be attributed to distinct chemical groups, oxidation generations, or chemical processes, but rather describe the average composition during specific time periods of the experiment."*

*"This could be a useful first-level simplification of the data, but suggests that PMF factors **derived from chamber experiments** cannot be used as surrogates for groups of reaction products within 3D models, because surrogate species should have chemical behavior that emulates real species."*

And in the conclusion:

*"We found that PMF **analysis of the chamber experiment described here** did not sort species into clear generations, since different species formed in a single generation can exhibit highly variable reactivities."*

(2) We agree with the reviewer that PMF and HCA have the same intended goals (at least in this work), but note that they are not mathematically similar. PMF is a matrix decomposition technique; no linear algebra is involved in HCA. The PMF algorithm attempts to minimize a clearly defined error function; HCA has no error function. The PMF decomposition produces factors that purportedly describe fundamental features of the data set, but which do not necessarily resemble the original measurements. HCA groups compounds together by similar time-series behavior, and the resulting groups must resemble their constituent measurements.

One of the fundamental features of PMF is that VOCs belong to multiple factors. Suppressing this feature so that PMF more resembles HCA defeats the purpose of the comparison: the mathematical differences between the different approaches result in a more, or less, scientifically meaningful reduction in complexity.

We appreciate the reviewer's comment about an "apples-to-apples" comparison. Now that we have more clearly presented in the introduction the goals of the data reduction, and the evaluation criteria, a more direct comparison of PMF and HCA (and GKP) is possible. The evaluation criteria are:

1) whether the resulting surrogates have chemically realistic behavior
2) whether the surrogates have the same range of chemical properties as the original data set
3) which subjective choices the researcher needs to make when implementing the method
4) what other new information about the system can be learned

The last paragraph in section 3.1.2 (discussion of PMF results), has been edited to read,

*"We conclude that in chamber experiments such as the one considered here, the PMF factors generally cannot be attributed to distinct chemical groups, oxidation generations, or chemical processes. Surrogate species derived from PMF factors do not have chemically realistic behavior or the same range of chemical properties as the original data set. The information about the system that can be determined from PMF factors is the average composition during specific time periods of the experiment. The researcher must subjectively choose the number of factors. These factors are not chemically robust and this should be considered when comparing PMF factors between oxidation experiments or chemical systems."*

The last paragraph in section 3.2.2 (discussion of HCA results) has been edited to read,

*"The surrogate species derived from HCA clusters have chemically realistic behavior, and have a similar range of chemical properties as the original data set. As with PMF, the choice of the number of clusters is subjective. In addition to defining surrogate species, HCA can be used to visualize the range of behavior and degree of similarity between all compounds in a data set. The clustering algorithm is thus a viable approach for describing a continuum of kinetic behavior and chemical properties."*

A final paragraph in section 3.3.3 (discussion of GKP results) has been added,

*"Surrogate species defined by GKP have by definition kinetically realistic behavior. The resulting groups of compounds have a range of chemical properties similar to that of the original data set, regardless of whether they are grouped using HCA or grouped by similar k and m. The method of grouping is subjective, as is the choice of number of clusters (if HCA is used) or the number of bins (if compounds are grouped by similar k and m). A particular strength of GKP is the resulting kinetic characterization of each compound. The effective rate constant and generation number provide new information that can be used to assess proposed mechanisms or to guide the reactive behavior of surrogate species in a model."*

(3) The PMF algorithm seeks to minimize the quality-of-fit parameter Q, which is the sum over elements of ((observed-reconstructed)/(error))^2:

$$Q = \sum_{i=1}^{m} \sum_{j=1}^{n} (e_{ij}/\sigma_{ij})^2$$ where $e_{ij}$ is the difference between the observation of ion $j$ at time $i$ and the PMF reconstruction, and $\sigma_{ij}$ is the standard deviation (noise) of that measurement. It is the relative signal-to-noise ratio ($e_{ij}/\sigma_{ij}$) of each ion that matters for the PMF fit: the ions with the highest signal-to-noise ratio are given the most weight in the fit, which are not necessarily the ions with highest signal overall. If each ion is multiplied by a normalizing factor, the respective errors $\sigma_{ij}$ must also be scaled; the signal-to-noise ratio remains the same, and the result of the PMF fit does not change.

We could create an artificial error matrix so that each compound has the same signal-to-noise ratio, and thereby give each compound equal weight in PMF analysis. However, this creates new problems. What artificial standard-deviation should be chosen for the whole dataset? It cannot be zero, and the results will change depending on the selected value. Additionally, very noisy, poorly-detected compounds would now have a disproportionate influence on the PMF solution.

We think it is more useful to the atmospheric science community to present the results of PMF as it is typically implemented. The differences between PMF and HCA are features that should be considered in an evaluation. This is especially important because our findings have implications for the interpretation of PMF of ambient data (i.e. "aged" factors in ambient air are not a linear combination of more and less aged air masses).

It would be helpful to make mirroring figures so that comparing the performance of each of the three techniques is easier. For instance, make figure 7 for PMF look like figure 9 for HCA.

Include a figure like figure 6b (mass spectrum of each factor) but for HCA results as part of figure 8. Same with GKP. Is the reason that PMF factors do not accurately represent chemical age groups is because each factor contains compounds with a wide range of amu (as shown in figure 6)? Is this not the case for HCA (please show in figure 8). And also for GKP.

We thank the reviewer for this very good suggestion.

Each technique has a graphical description that is unique to that particular approach. For PMF, this is the comparison between the measured and reconstructed total signal, which is currently Figure 4c (previously Figure 6) . For HCA, this is the dendrogram, which is currently Figure 7a (previously Figure 8). For GKP, this is the best fit of the parameterization to the measurements, which is currently shown in Figure 10 (previously Figure 11). We kept these figures, because they are necessary for the discussion in each respective section.

For the surrogate species derived from each approach, there are four graphical descriptions that can be directly compared: the time series, the mass spectra, the plot of $k$ vs $m$, and the oxidation state plot (O:C vs num C). We decided that it would be best to show all of these in a single figure at the end of the paper. The previous Figure 14 shows some of this information, but is missing the PMF results, time series, and mass spectra.

We have revised the paper as follows:

- A new section is added at the end of Results and Discussion: Section 3.4, Comparison of PMF, HCA, and GKP.
- The new section 3.4 includes a revised Figure 13 (a revision of previous Figure 14), and a paragraph discussing the figure.

The new Figure 13 shows the $k$ vs $m$ plot, the oxidation state plot, the time series of several surrogates, and the mass spectra of several surrogates, for PMF, HCA, and GKP.

There are several possible ways to use PMF, HCA, and GKP to group compounds, that result in any number of groups. For example, PMF solutions were determined for one to ten factors. If we were to include all these possibilities in Figure 13, the figure would be unreadably complex. Therefore, we chose to show just one possible grouping for each technique. For PMF, we chose to show the six-factor solution; for HCA, we chose to show the grouping where the precursor is separated from all product species; for GKP, we show two possible ways of grouping compounds, one using HCA, and the other using binning by $k$ vs $m.$ These choices of groupings are discussed the most extensively in the text.

We also restricted the number of time series and mass spectra shown in Figure 13. The reason for this is legibility. The HCA solution has nine clusters containing two or more compounds and 10 with just one species. The GKP solutions have a similarly large number of groups. A figure with twenty time series in one panel isn't readable. Because

the PMF solution has 6 factors, we show the 6 groups with the highest total carbon concentration from the HCA and GKP solutions. The six largest groups account for about 80% of the total carbon concentration in products in each case.

The new Figure 13 and caption are shown below.

[Figure]

Figure 13 Overall comparison of groups derived from PMF, HCA, and GKP of chamber data. The columns show, from left to right, the results of A. PMF, B. HCA, C. GKP best-fits grouped using HCA, and D. measurements grouped by GKP fit parameters. The rows show, from top to bottom, 1. the average carbon oxidation state and number of carbon atoms per molecule for each group, 2. the time series of the six groups containing the most carbon, 3. the mass spectra of those six groups, and 4. the rate constant and generation number of each group. Within each column, each chemical group is assigned a specific color. This color scheme is the same for each plot within a column. The marker area is proportional to the averaged concentration (parts-per-billion carbon) of all species in the group, with the marker size of the precursor (red) decreased by a factor of 2 for legibility. The marker area scheme is consistent across all plots. PMF factors do not have kinetically realistic time series, therefore there is no plot A4.

The text in section 3.3.3 (clustering of GKP) was edited to be consistent with the new location and content of Figure 13:

*"Results from each approach, showing both kinetic characteristics (k and m) and chemical properties (oxidation state and carbon number) of each group, are given in Figure 13, which includes an overview and comparison of grouped species derived from PMF (Figure 13a), HCA (Figure 13b), and GKP (Figure 13c and d)."*

Discussion was added to section 3.4, Comparison of PMF, HCA, and GKP:

*"In all cases, the majority of the carbon can be represented by a manageable number of groups. The relationship between oxidation state and number of carbon per molecule is similar, regardless of the grouping technique. The PMF factors have a smaller range of chemical properties than chemical groupings derived from HCA or GKP. The range of chemical properties is similar for HCA and GKP. The time-series of PMF factors are clearly different from those of HCA- and GKP-derived groups, and have non-kinetically-realistic shapes with sharp maxima.*

*The PMF factors each contain many more compounds than the groups derived from HCA or GKP. Many of the same compounds are consistently grouped together by HCA and GKP, regardless of whether HCA, HCA of GKP, or binning of GKP is used. Additionally, the range of kinetic properties, and the locations of major compound groups in kinetic space, are similar between the HCA and GKP approaches. This reproducibility suggests that these are chemically meaningful compound groupings. Some groups derived from HCA or GKP contain only a single species. These could be chemically important compounds whose unique behavior should be considered when modeling the system; conversely, they could be measurement outliers which should be discarded. The interpretation of these species is subjective.*

*Regardless, the combination of fitting using the GKP and grouping based on kinetic behavior may provide a viable approach for greatly simplifying the time-dependent behavior of complex mixtures of reaction products in a laboratory oxidation system."*

The way that HCA is described on page 12, it reads as if the technique also solves for the final number of clusters needed to explain the variance of all input members. But it turns out (page 23 line 465) that this final cluster number is "chosen" by the user. How was this final number objectively determined? There are eight clusters (figure 8), but that can easily be reduced further (6, 7 and 8 look pretty similar, as do 3 and 4; and conversely, each of the those eight can be split even further). PMF at most has 6 factors. GKP has 9. How are the final group/factor/cluster number chosen for PMF and GKP? Perhaps the authors should choose the same number for the three techniques. This again, I think will present a fairer comparison of the three techniques.

When using PMF, HCA, or GKP, the number of factors, clusters, or groups is subjectively chosen by the researcher. The method to appropriately determine the number of clusters is of course different depending if PMF, HCA, or GKP is used. We did consider (and have discussed in the manuscript) how different numbers of factors or groups affect the

interpretation of the data. For PMF, we considered solutions with one to ten factors. For HCA, we considered a range of threshold values to define clusters with distinctly different behavior. Results from groupings with 13 clusters (of which 5 have significant intensity) to 120 clusters (13 significant clusters) are presented. In section 3.3.2, we extensively discuss different ways to group compounds using GKP, each of which results in a different number of groups.

To clarify and expand this discussion, we have edited the text as follows:

At the end of Section 2.2.1 (implementation of PMF), we inserted,

*"When PMF is used to reduce the complexity of a dataset, the number of factors must be chosen by the researcher, a choice that is inherently subjective. Solutions were explored with one to ten factors for the synthetic dataset and the chamber data."*

At the end of section 2.2.2 (implementation of HCA), we inserted,

*"Compounds must be grouped into a specific number of clusters in order to use HCA to define surrogate species. The average chemical and kinetic properties of each cluster can be used to define a surrogate species. As with the number of factors from PMF, the number of clusters is subjectively chosen by the researcher. The clusters could be selected by hand, or by choosing a threshold for distance dAB to automatically define clusters. We chose to use a threshold to define the number of clusters, and considered several different values of thresholds that result in different numbers of clusters. The effect of threshold value on the interpretation of the data is discussed in Section 3.2."*

At the end of section 2.2.3 (implementation of GKP), we inserted,

*"Compounds can be grouped by similar k and m to reduce the complexity of the dataset. The k, m, and average chemical properties of the group can be used to define a surrogate species. The choice of the number of groups and the method of grouping are subjective. GKP could be used alone, by binning compounds by similar k and m, or it could be used in combination with another analysis technique, such as HCA. Several approaches to using GKP to define surrogate species are discussed in section 3.3.2."*

In the overall comparison of all techniques, Figure 13, we show the six groups with the highest carbon concentration derived from each technique.

Finally, in section 4 (conclusion) at line 723, we added,

*"All three approaches require a subjective choice of the number of compound groups."*

There are sections dedicated to PMF (section 3.1) conducted on both simulated and chamber data. Same with GKP (section 3.3). Why not HCA (section 3.2)?

We have inserted a dedicated section for HCA of simulated data (Section 3.2.1). We moved the original Figure 3 (now Figure 6) to this section. This figure shows HCA applied to simulated data

and was originally used as a visual explanation of the algorithm. The description of the figure in the text was also moved to this new section.

We added the following discussion to section 3.2.1:

*"In this example with simulated data, HCA generally clusters together compounds of similar generation, though not perfectly. HCA clusters together compounds that have similar time-series behavior, and time-series behavior is determined not only by generation, but also by formation and reaction rate constants. For example, species B1, B2, and C2 all have fast formation and reaction rates, resulting in similar time-series. HCA groups these three species together. The algorithm suggests further that the first-generation products B1 and B2 are much more similar to one another, than they are to second-generation product C2.*

*The results of HCA applied to synthetic data indicate several strengths and weaknesses of the HCA algorithm. Most importantly, the algorithm provides a clear way to visualize the behavior and relationships between all measurements in a dataset. The precursor compound can be included in the analysis, because data are normalized and the high intensity of the precursor does not skew the results. Compounds with similar kinetic properties are mostly grouped together, but some generational miscategorization still occurs. It may be difficult to use HCA to separate compounds which have different generation numbers but similar formation and reaction rates.*

*HCA can be used to simplify the dataset, by replacing clusters of compounds with surrogates. If the surrogate time-series behavior is determined by averaging the time-series of the individual members of the cluster, then the surrogate will have chemically realistic behavior. As noted previously, the researcher must subjectively choose the number of clusters."*

We added the following discussion to section 3.2.2 (HCA of chamber data):

*"There are some significant differences between the synthetic data set, and real-world data sets collected from chamber experiments. Most importantly, the actual chamber experiment includes many more species (ten species in the synthetic system, compared to thousands of detected ion masses and hundreds of measured species in the chamber experiment). The real chamber data set includes many non-meaningful measurements whose time-series have no structure. Additionally, many species in the real-world data set have much more similar time-series behavior to one another than any two of the species in the synthetic system. Conversely, there are also distinct outliers in the real-world data set, whose time-series behavior does not resemble any other compound. HCA effectively separates meaningful from non-meaningful measurements, groups together very similar compounds, and highlights outliers."*

Minor Figure 1 bottom. legends needed.

This has been corrected.

page 7, line 167: "Teflon" itself is PTFE and is trademarked and manufactured by a company called Chemours. Is the tubing PFA or PTFE? Manufactured by Chemours? If not, it is not Teflon.

The tubing is PFA; the name has been corrected, and similarly in section 2.1.2 (Teflon chamber to PFA chamber).

Figure 2. Panel D. Not a great figure to highlight in main manuscript. Account for oscillation before including as main figure. I understand it is not included, but most people only look at figure and not read caption, will come away with wrong impression. Panel B not informative. This figure perhaps is introduced prematurely since hierarchical clustering since that section is far below.

Both reviewers found this figure distracting and complicated. We moved Figure 2 to the supplemental information. Other researchers who are working with CIMS data may find it a useful guide.

Page 12 line 290, Need hyphen in citation

This has been corrected.

Figure 8, 9 and 14 share the same color scheme. It would be nice to have a common legend and/or color-bar shown in each of these figures to remind the reader that these colors represent generation determined by HCA.

The color in original Figures 8,9, and 14 (now 7,8, and 13) is only used to distinguish clusters, and does not have anything to do with generation. The coloring in all figures that show simulated data is consistent (precursor in black, 1st generation in red, 2nd generation in yellow, and 3rd generation in blue). We edited the figure captions to clarify the use of color in all figures.

Please make clear in each of the figure caption in the SI and main manuscript whether it is presenting simulated data or measured data.

The figure captions have been edited according to the reviewer's suggestion.

**Anonymous Referee #2**

The authors compare three different techniques for dimensionality reduction in mass spectrometry time series data sets, positive matrix factorization, hierarchical clustering, and gamma kinetics parameterization. They evaluate the behaviour of the three techniques on two data sets and conclude that PMF is not competitive compared to the two others.

Overall the paper gives a good overview of the work, but requires some revision before being published.

For the clustering, the authors chose agglomerative clustering using average linkage. However, they don't provide all necessary information to reproduce the experiments or motivate their choices. Euclidean distance was used, why was this distance measure chosen? I can see it

makes sense in some ways, however there are different metrics specific for time series, most notably dynamic time warping (DTW), which might make sense in this case. DTW compensates for shifts in the time series, so for particular use cases, this could make sense. Otherwise, there are more approaches to achieve a clustering, why choose this one? Agglomerative tends to be computationally faster than divisive, but this shouldn't be a problem with data sets this size. Otherwise what about density-based clustering or really simple approaches such as K-Means?

Here we address three points raised by the reviewer.

(1) Why was Euclidean distance used?

Simply put, we tried several distance measures and found that Euclidean distance resulted in the grouping that was most consistent, understandable, and insensitive to outlier points in time-series data.

We added this to the text at line 306,

*"Other distance metrics are possible, including using a correlation coefficient or the sum of squared residuals. This particular approach was chosen because it resulted in the grouping that was most reproducible and understandable, and least sensitive to outlier points in the time series."*

(2) Why was HCA used instead of some other clustering method?

When we began this work, we did try several other approaches to clustering, including K-means. K-means is suited for data sets where there are several distinct, discrete groups. It also doesn't handle outliers well: all compounds have to be assigned to a cluster. Density-based clustering is similar to K-means in that it identifies discrete groups of compounds; the implementation is a little different, and it is somewhat better equipped to handle outliers and strangely-shaped clusters.

We found that K-means did not work well with this particular data set. In the chamber data set described here, each sample is one chemical species, the variables are time t1, t2, t3, etc. and the vector description of each sample is the normalized intensity at each of ~500 time points. In this implementation k-means attempts to find clusters in 500-dimensional space. It is really not the optimal technique to do this, especially since there is a comparatively small number of samples (about 500 compounds). Additionally, the data describes a continuum of behavior, rather than distinct, discrete groups.

We found HCA to be a method ideal for this type of data, which is why it is featured in the final manuscript. It is easy to implement with time-series data, it returns not just groupings of compounds, but also a metric of the similarity of behavior, it can describe a continuum of behavior, and it handles outliers well.

(3) Suggestion of dynamic time warping.

This is a very interesting suggestion and could work well when combined with GKP. With this implementation it may be possible to remove the effect of different rate constants and group species by generation alone. More work is needed to explore this approach.

Similarly, there exist a wide range of algorithms for PMF, which one was actually used here? And why this one? The authors give the library, but some details on the method would be necessary.

We used the PMF Evaluation Tool v2.08. It is based off the PMF2 algorithm from Paatero 2007. This particular implementation is widely used in atmospheric science and therefore the evaluation of this particular technique is likely of the greatest interest to the atmospheric science community. We edited the text in section 2.2.1 (implementation of PMF) to read,

*"The algorithm was implemented using the PMF Evaluation Tool v2.08 (Ulbrich et al., 2009) using the PMF2 algorithm (Paatero, 2007). We chose this implementation because it is widely used in atmospheric science and has been optimized for atmospheric chemistry data."*

The algorithm is a constrained least-squares approach and is already described in the text.

On page 11, the authors give a formula for the quality of fit parameter Q, but half of the variables in the formula are not defined anywhere so the formula does not really make sense.

This section now reads,

*"Briefly, the algorithm takes as input an m×n matrix of measured data M, **containing n measured compounds at m time points**, and a matrix of estimated error (one standard deviation, σ) for each point in the measured data matrix. The solution for a given number of factors p is given as an m×p matrix G of factor time series, a p×n matrix F of factor profiles, and a matrix E that contains the residual (M-GF). F and G are iteratively adjusted to minimize the quality-of-fit parameter Q:*

$$Q = \sum_{i=1}^{m} \sum_{j=1}^{n} (e_{ij}/\sigma_{ij})^2$$

*where eij is the residual between the measurement and the PMF reconstruction of compound j at time point i, and $\sigma_{ij}$ is the estimated error of that measurement."*

Also, when looking at algorithms such as PMF or clustering, it would be interesting to calculate performance measures and give them to get a feeling how well the clustering or factorization works. This could be simple reconstruction error or normalized mutual information (if there is a ground truth).

The PMF algorithm includes a reconstruction error. In the text we call this the "residual" and state that for the chamber experiment it is quite low, about 2%, regardless of aging time. It is consistently low for solutions with three or more factors. At line 410 we edited the text to note that "residual" may also be called "reconstruction error". Despite the low residual, the PMF factors do not seem to be a chemically realistic deconstruction of the data set.

The HCA algorithm does not lend itself to a reconstruction error. Unlike PMF, the algorithm does not seek to minimize an error term. Each species is assigned to a single cluster, and the time-series of clusters necessarily resembles the individual cluster contributors. Normalized mutual information can provide a way to assess the quality of clustering. Unfortunately, we do not know the exact chemical identities and mechanistic relationships of all 464 compounds measured during the chamber experiment, so it isn't possible to use NMI to assess the clustering as a whole. However, we can calculate NMI for the HCA of the synthetic system. We calculated NMI for the synthetic system for solutions with one to ten clusters, evaluating the separation of different generations into distinct clusters. We also calculated NMI for the PMF solutions with 2 to 10 factors. We used the relative intensities of each generation in each factor to calculate NMI. For example, if Factor 2 accounted for 40% of the total integrated intensity of B1 and 80% of the total intensity of B2, we assigned a value of 1.2 for Generation B to Factor 2. The results are:

| Number of clusters or factors | PMF NMI | HCA NMI |
|---|---|---|
| 2 | 0.402056 | 0.396705 |
| 3 | 0.380505 | 0.466626 |
| 4 | 0.436106 | 0.520791 |
| 5 | 0.42733 | 0.682508 |
| 6 | 0.441523 | 0.744987 |
| 7 | 0.761499 | 0.834656 |
| 8 | 0.733057 | 0.799452 |
| 9 | 0.678975 | 0.755557 |
| 10 | 0 | 0 |

HCA has a higher NMI value in each case, except for when only 2 factors/clusters are chosen. In this case, the PMF residual is high (13%).

We have included this information in the text. Section 3.1.1. (PMF of synthetic data) now reads:

*"A set of PMF solutions for the synthetic data, including 2-10 factors, is shown in the Supplement (Figure S5). The quality of the PMF reconstruction can be evaluated in two ways: the residual between the PMF reconstruction and the original data (lower residual indicates better agreement), and the normalized mutual information (NMI) (Vinh et al., 2010) between PMF factors and photochemical generation. The PMF residual is high for the 2-factor solution (13%, on average), and low for 3- to 10-factor solutions (less than 5%).*

*The normalized mutual information metric describes the correlation between PMF factors and generation. A value of 0 means no correlation, and a value of 1 indicates that generations are perfectly assigned to distinct factors. Because species can be assigned to multiple factors, we used the relative intensities of each generation in each factor as input to the NMI calculation. For instance, if PMF Factor 2 accounted for 66% of the total integrated intensity of first-generation product B1, 97% of the intensity of B2, and 12% of the intensity of B3, we assigned a value of 1.75 for first-generation products to Factor 2. The mutual information describes the probability that products of a particular generation are assigned to the same*

*cluster. Mutual information must be normalized so that it can be compared between solutions with different numbers of factors or clusters. As the normalization factor, we used the arithmetic average of the generation and factor entropy, which is a quantity that describes the size and diversity of values in the two classification schemes (generation and PMF factor).*

*NMI values are provided in Table 1. For purposes of comparison, Table 1 also includes the NMI values calculated from hierarchical clustering analysis. HCA of the synthetic data set is described in section 3.2.1. Because there are only ten species in the synthetic data set, a solution with ten groups, each of which contains a single species, has no correlation between generation and groups, and the NMI is zero.*

| Number of groups (PMF factors or HCA clusters) | PMF NMI | HCA NMI |
|---|---|---|
| 2 | 0.402 | 0.397 |
| 3 | 0.381 | 0.467 |
| 4 | 0.436 | 0.521 |
| 5 | 0.427 | 0.683 |
| 6 | 0.442 | 0.745 |
| 7 | 0.761 | 0.835 |
| 8 | 0.733 | 0.799 |
| 9 | 0.679 | 0.756 |
| 10 | 0 | 0 |

*Table 1. Synthetic data. Normalized mutual information index quantifying the correlation between PMF factor and photochemical generation.*

*Figure 3 shows the four-factor solution. The four PMF factors are able to reconstruct the total signal with excellent agreement, but they do not correspond to the four original generations of compounds (precursor plus three product generations). There is some relationship between early, middle, and late-generation species and the PMF factors **(indicated by non-zero NMI values)**, but regardless of the selected rotational forcing, all PMF factors contain species from more than one generation. For instance, because both C1 and D2 are long-lived species, they are correlated over the time period of the experiment and so are assigned to the same factor. More importantly, many species are included in two or more PMF factors, despite being formed by only one pathway. Eight to ten factors (approximately the number of species in the dataset) are needed to separate generations, which is not a useful simplification of the data set (which is made up of only ten species)."*

Section 3.2.1 (HCA of synthetic data) reads:

*"The ability of HCA to separate compounds of different generations was quantified by the normalized mutual information (NMI). NMI values are provided in Table 1. For all solutions with more than 2 clusters (or factors), NMI values for HCA are higher than those of PMF, indicating that HCA more successfully sorts compounds by generation."*

Another issue is repetition of experiments. While the agglomerative clustering should be mostly stable, PMF usually is not when using big enough data sets. So a single run would not be a reliable representation and multiple runs would be necessary. Additionally, this paper seems to base its results on two data sets, which cannot give any reliable or statistically sound

performance representation for these approaches. Anything below at least 5 data sets won't give you the proof you need for what you state in the conclusion. Either rerun the experiments a lot more times or restate in the conclusion that this gives an indication, but to proof it, many more experiments would be needed.

The data in this paper were provided by a specialized instrumentation suite that was only available for a short period of time, so unfortunately we are not able to re-run all the experiments. However, we argue that the results of this work are meaningful despite the low number of data sets used.

First, in atmospheric chemistry, PMF is often run on single data sets, simply because it is not possible to re-produce conditions in the ambient atmosphere. Nonetheless, PMF has been extremely valuable to the atmospheric science community, and the results are often clearly meaningful, even when only a single data set is used.

Second, we assessed the stability of the PMF solution by running PMF several times with different random seed values. We found that the PMF solutions did not significantly change with seed value. We note this in section 2.2.2.

Third, we make no definitive statements about what the various techniques can or cannot be used for, that would depend on analysis of many data sets. In our assessment of PMF, we claim that the factors do not necessarily represent distinct chemical groups. We suggest HCA as a method to organize data in chamber experiments, and present GKP as a method to derive kinetic information from a chamber experiment. All of these statements can be (and are) clearly shown by one or two systems.

To clarify the limitations of our data sets, we have added in the conclusion (line 753, in discussion of future work),

*"The current analysis is based on two systems, a synthetic system and a chamber experiment, and more work is needed to see how these analysis approaches perform with other systems."*

I would recommend improving the presentation, particularly the figures. It is not always straight forward to understand what is shown. For example Figure 14 on its own does not explain the meaning of the colours or the size of the dots. Also the labeling of part A, B, and C is not very standard and sometimes hard to understand.

We made minor edits to the figure captions to always explain the color scheme and marker size. We made minor cosmetic changes to a few figures to improve legibility and colorblind-friendliness. Labels were added to Figure 3 (previously Figure 5), lines were thickened in Figure 10 (previously Figure 11), and the color scheme was changed slightly in Figure 12 (previously Figure 13). Multi-panel figures are always labeled A, B, C, D from left to right, top to bottom. The panels A B C etc. are described in the caption of each figure.

One specific question I have on Figure 2. I don't see how the cluster of C happens,looking at the data in C, this does not seem to be a cluster, the highest gray line is faraway from all others and

looks far too much as an outlier compared to the rest of the cluster. How does this compare to the rest of the data? Is everything else just evenfurther away?

Both reviewers found this figure distracting and complicated. We moved Figure 2 to the supplemental information. Other researchers who are working with CIMS data may find it a useful guide.